# 14-3-3ζ regulates the mitochondrial respiratory reserve linked to platelet phosphatidylserine exposure and procoagulant function

Simone M. Schoenwaelder[1,2,3,*], Roxane Darbousset[1,2,3,*], Susan L. Cranmer[3,*], Hayley S. Ramshaw[4,*], Stephanie L. Orive[3], Sharelle Sturgeon[3], Yuping Yuan[1,2,3], Yu Yao[3], James R. Krycer[2,5], Joanna Woodcock[4], Jessica Maclean[1,2,3], Stuart Pitson[4], Zhaohua Zheng[3], Darren C. Henstridge[6], Dianne van der Wal[3], Elizabeth E. Gardiner[3,7], Michael C. Berndt[8], Robert K. Andrews[3], David E. James[2,5,9], Angel F. Lopez[4] & Shaun P. Jackson[1,2,3,10]

The 14-3-3 family of adaptor proteins regulate diverse cellular functions including cell proliferation, metabolism, adhesion and apoptosis. Platelets express numerous 14-3-3 isoforms, including 14-3-3ζ, which has previously been implicated in regulating GPIbα function. Here we show an important role for 14-3-3ζ in regulating arterial thrombosis. Interestingly, this thrombosis defect is not related to alterations in von Willebrand factor (VWF)–GPIb adhesive function or platelet activation, but instead associated with reduced platelet phosphatidylserine (PS) exposure and procoagulant function. Decreased PS exposure in 14-3-3ζ-deficient platelets is associated with more sustained levels of metabolic ATP and increased mitochondrial respiratory reserve, independent of alterations in cytosolic calcium flux. Reduced platelet PS exposure in 14-3-3ζ-deficient mice does not increase bleeding risk, but results in decreased thrombin generation and protection from pulmonary embolism, leading to prolonged survival. Our studies define an important role for 14-3-3ζ in regulating platelet bioenergetics, leading to decreased platelet PS exposure and procoagulant function.

---

[1] Heart Research Institute, Thrombosis Group, Newtown, New South Wales 2042, Australia. [2] Charles Perkins Centre, Cardiovascular Division, The University of Sydney, Camperdown, New South Wales 2006, Australia. [3] Australian Centre for Blood Diseases, Alfred Medical Research and Education Precinct (AMREP), Monash University, Prahran, Victoria 3004, Australia. [4] Centre for Cancer Biology, SA Pathology and the University of South Australia, Adelaide, South Australia 5000, Australia. [5] School of Life & Environmental Sciences, The University of Sydney, Camperdown, New South Wales 2006, Australia. [6] Baker IDI Heart and Diabetes Institute, Melbourne, Victoria 3004, Australia. [7] ACRF Department of Cancer Biology and Therapeutics, John Curtin School of Medical Research Australian National University, Canberra ACT 2600, Australia. [8] Faculty of Health Sciences, Curtin University, Perth, Western Australia 6102, Australia. [9] Sydney Medical School, The University of Sydney, Camperdown, New South Wales 2006, Australia. [10] Department of Molecular and Experimental Medicine, The Scripps Research Institute, La Jolla, California 92037, USA. * These authors contributed equally to this work. Correspondence and requests for materials should be addressed to S.P.J. (email: shaun.jackson@sydney.edu.au).

The 14-3-3 family of adaptor proteins are ubiquitously expressed in eukaryotic cells and were the first identified phosphoserine- and phosphothreonine-binding proteins. They interact with a wide variety of intracellular molecules, including signalling proteins, metabolic enzymes, cytoskeletal proteins, transcription factors, apoptosis regulators and tumour suppressor proteins[1,2]. Consequently, they play a key role in regulating cell cycle progression, apoptosis, metabolism, intracellular trafficking and in the regulation of cell stress responses[1,3].

Seven 14-3-3 isoforms have been identified in mammalian cells: β, ε, ζ, γ, η, τ and σ. These isoforms associate as homodimers or heterodimers to form a negatively charged channel that constitutes a highly conserved ligand-binding pocket. The 14-3-3 molecules can bind more than 200 different proteins, with each 14-3-3 isoform showing a distinct preference for selected proteins. The interaction with 14-3-3 can have a major effect on target protein function, altering their localization, stability, conformation and/or phosphorylation[4].

14-3-3 proteins have been extensively investigated in nucleated cells, however these proteins are also expressed in high levels in anucleate cells including platelets, raising the possibility of additional roles unrelated to gene transcription and proliferation. Six 14-3-3 isoforms have been detected in human platelets including β, ε, ζ, γ, η and τ, with ζ and γ expressed at high levels. The first identified 14-3-3 binding partner in platelets was the VWF receptor GPIb/V/IX, an adhesive molecule with a key role mediating platelet–vessel wall and platelet–platelet adhesion under conditions of high shear stress[5–7]. 14-3-3ζ binds to the cytoplasmic tails of GPIbα and GPIbβ (ref. 8) and numerous studies have focussed on the functional relevance of this interaction for platelet activation and signalling. Some studies have suggested that the GPIb-14-3-3ζ interaction can promote VWF-dependent integrin $\alpha_{IIb}\beta_3$-activation and cell spreading[9,10], whereas others have proposed a negative signalling role[11,12]. It has been suggested that 14-3-3ζ binding to GPIbα might regulate the ligand binding function of the receptor[13], although several other studies have found no role for the 14-3-3ζ binding domain of GPIbα in regulating GPIbα adhesive function[9,14–17]. Understanding the role of the GPIb-14-3-3ζ interaction in regulating platelet adhesion and activation is complicated by the finding that all known isoforms of 14-3-3 in platelets can bind to the cytoplasmic tail of GPIbα (ref. 18), making interpretation of the role of a single 14-3-3 isoform difficult.

In the current study, we have investigated the importance of 14-3-3ζ in regulating platelet function by studying mice genetically deficient in this 14-3-3 isoform[19]. These studies have uncovered a non-redundant role for 14-3-3ζ in regulating the prothrombotic function of platelets in vivo. Interestingly, this thrombosis defect is not due to abnormal VWF–GPIb adhesive function, nor defective platelet activation induced by soluble platelet agonists, but rather, through reduced platelet PS exposure and procoagulant function. This defect in procoagulant function in 14-3-3ζ-deficient mice is associated with reduced arterial thrombosis, as well as reduced thrombin generation and pulmonary embolism in vivo. Analysis of platelet bioenergetics has revealed enhanced mitochondrial respiratory reserve capacity in 14-3-3ζ-deficient platelets that correlated with sustained levels of metabolic ATP. These studies identify 14-3-3ζ as an important regulator of platelet bioenergetic function, linked to platelet procoagulant function and thrombosis in vivo. Targeting platelet metabolic pathways may represent a new approach to antithrombotic therapy.

## Results

### 14-3-3ζ-deficient mice are protected from arterial thrombosis.
To investigate the importance of 14-3-3ζ in promoting thrombosis in vivo, we utilized mice in which the 14-3-3ζ gene (Ywhaz) had been knocked out globally using Lexicon gene trap vector technology, as described previously[19]. We initially compared the ability of these mice, referred to herein as 14-3-3ζ-deficient mice, and matched 14-3-3ζ litter mate controls to form occlusive arterial thrombi following electrolytic injury to carotid arteries[20]. Electrolytic injury initiates the formation of platelet-rich thrombi that typically occlude the artery within 15–20 min (ref. 20). Analysis of carotid blood flow following injury revealed a significant decrease in the ability of 14-3-3ζ-deficient mice to form occlusive thrombi (Fig. 1a–e), with stable vessel occlusion occurring in 80% of 14-3-3ζ-wild-type (wt) mice compared with 20% of 14-3-3ζ-deficient mice (Fig. 1e). Consistent with this, vessel patency time in 14-3-3ζ-deficient mice was ∼5-fold longer than for 14-3-3ζ-wt mice (Fig. 1c), and time to initial vascular occlusion was significantly delayed in 14-3-3ζ-deficient mice (Fig. 1d). These studies suggest an important role for 14-3-3ζ in regulating arterial thrombus formation in mice.

14-3-3 proteins are ubiquitously expressed, therefore studies were performed on a cohort of bone marrow transplanted mice to confirm that the thrombosis defect was related to deficiency of 14-3-3ζ in hemopoietic cells, rather than loss of 14-3-3ζ function in the vessel wall. Four independent cohorts of bone marrow-transplanted mice were generated, either on a 14-3-3ζ-wt or 14-3-3ζ-deficient background, and transplanted with either 14-3-3ζ-wt or 14-3-3ζ-deficient bone marrow. Mice from each cohort were challenged with electrolytic injury and carotid blood flow monitored for the formation of an occlusive thrombus. As demonstrated in Fig. 2a–d, the data obtained from these cohorts recapitulated the observations with pure 14-3-3ζ-wt or 14-3-3ζ-deficient mice (Fig. 1). Time to initial occlusion and vessel patency time was significantly increased in mice receiving 14-3-3ζ-deficient bone marrow relative to mice transplanted with 14-3-3ζ-wt bone marrow, regardless of genetic background (Fig. 2c,d). These studies confirm that 14-3-3ζ expression in hematopoietic cells (most likely platelets) is important for vascular occlusion by arterial thrombi. In control studies, a slight reduction in the circulating platelet count was observed in 14-3-3ζ-deficient mice (14-3-3ζ-wt mean = 1,108 ± 57; 14-3-3ζ-deficient mean = 963 ± 39), however this was not sufficient to cause a thrombosis defect in mice[21]. This mild decrease in the platelet count was unlikely to be due to increased platelet apoptosis and clearance as the lifespan of 14-3-3ζ-deficient platelets was identical to 14-3-3ζ-wt controls (Supplementary Fig. 1a). In further control studies, we confirmed that there was no altered expression of the major platelet surface receptors on the surface of 14-3-3ζ-wt versus 14-3-3ζ-deficient platelets (Supplementary Fig. 1b).

To examine whether deficiency of 14-3-3ζ results in a bleeding phenotype, we examined the haemostatic response in 14-3-3ζ-deficient and 14-3-3ζ-wt mice following surgical resection of the mouse tail[22]. Examination of bleeding time using a 3 mm tail lop revealed minimal impact on the haemostatic response (Fig. 2e), with a non-significant increase in the time to cessation of bleeding. Furthermore, there was no evidence of spontaneous or surgical bleeding in any of the mice examined. Overall, these studies suggest an important role for 14-3-3ζ in regulating the prothrombotic function of platelets, yet is functionally redundant for the platelet haemostatic response.

### Normal activation and adhesion in 14-3-3ζ-deficient platelets.
To gain insight into the mechanisms by which 14-3-3ζ regulates thrombus formation in vivo, we performed in vitro flow studies on a fibrillar Type I collagen substrate using

anticoagulated whole blood. The dynamics of platelet adhesion and aggregation were examined in real-time and revealed no major differences in thrombus formation between 14-3-3ζ-wt and 14-3-3ζ-deficient mice (Fig. 3a). Moreover, VWF binding to GPIbα (Fig. 3b), VWF–GPIbα-mediated platelet aggregation (Fig. 3c) and platelet adhesion to immobilized VWF (Supplementary Fig. 2a) was normal in mouse platelets deficient in 14-3-3ζ. Similarly, integrin $\alpha_{IIb}\beta_3$-dependent platelet aggregation (Fig. 3d; Supplementary Fig. 2b) and agonist-induced platelet activation (P-selectin expression and integrin $\alpha_{IIb}\beta_3$ activation, Fig. 3e,f), were also unperturbed in 14-3-3ζ-deficient platelets, or human platelets pretreated with a 14-3-3 dimer destabilizer[23] (Supplementary Fig. 3a,b). Multiple 14-3-3 isoforms are expressed in platelets (Fig. 3g), all of which can associate with GPIbα (ref. 18). In control studies, we confirmed by immunoprecipitation and immunoblot analysis that 14-3-3ζ associates with GPIbα in 14-3-3ζ-wt platelet lysates (Fig. 3h). We also confirmed the loss of 14-3-3ζ in 14-3-3ζ-deficient platelet lysates by immunoblot analysis (Fig. 3g). To determine the extent of 14-3-3 binding to GPIbα in 14-3-3ζ-deficient platelets, GPIbα was immunoprecipitated from whole-cell platelet lysates and the immunoprecipitates blotted with a pan-14-3-3 polyclonal antibody (Fig. 3i). Similar levels of 14-3-3 binding to GPIbα was observed in 14-3-3ζ-wt and 14-3-3ζ-deficient platelets, suggesting that other 14-3-3 isoforms compensate for the loss of 14-3-3ζ.

**Reduced procoagulant function in 14-3-3ζ-deficient platelets.** An important function of activated platelets is their ability to support the assembly of coagulation complexes on their plasma membrane, a process requiring PS exposure necessary for localized α-thrombin generation and fibrin formation. Both thrombin stimulation of platelets and fibrin generation are important for thrombus growth and stability, and as a consequence, drugs that target thrombin's actions inhibit arterial thrombus growth[24]. To examine whether platelet procoagulant function was affected in 14-3-3ζ-deficient mice, we examined exposure of PS on the surface of thrombin and collagen-related peptide (CRP)-stimulated platelets. This agonist combination induces high, sustained cytosolic calcium flux and platelet PS exposure through a cell death pathway linked to mitochondrial-regulated cell necrosis[25,26]. In response to co-stimulation with thrombin/CRP, the ability of 14-3-3ζ-deficient platelets to externalize PS, as detected through Alexa 488-labelled Annexin V-binding, was significantly impaired (Fig. 4a), with an identical reduction observed using FITC-lactadherin as an alternative probe for PS. Similarly, pre-treating human (Fig. 4b) or 14-3-3ζ-wt mouse platelets (Fig. 4c) with a 14-3-3 dimer destabilizer (RB-011)[23] resulted in a specific defect in agonist-induced PS exposure. A similar reduction in the development of procoagulant platelets was apparent in RB-011-treated platelets adherent to an immobilized collagen substrate under flow conditions (Supplementary Fig. 3c). Consistent with a reduction in PS

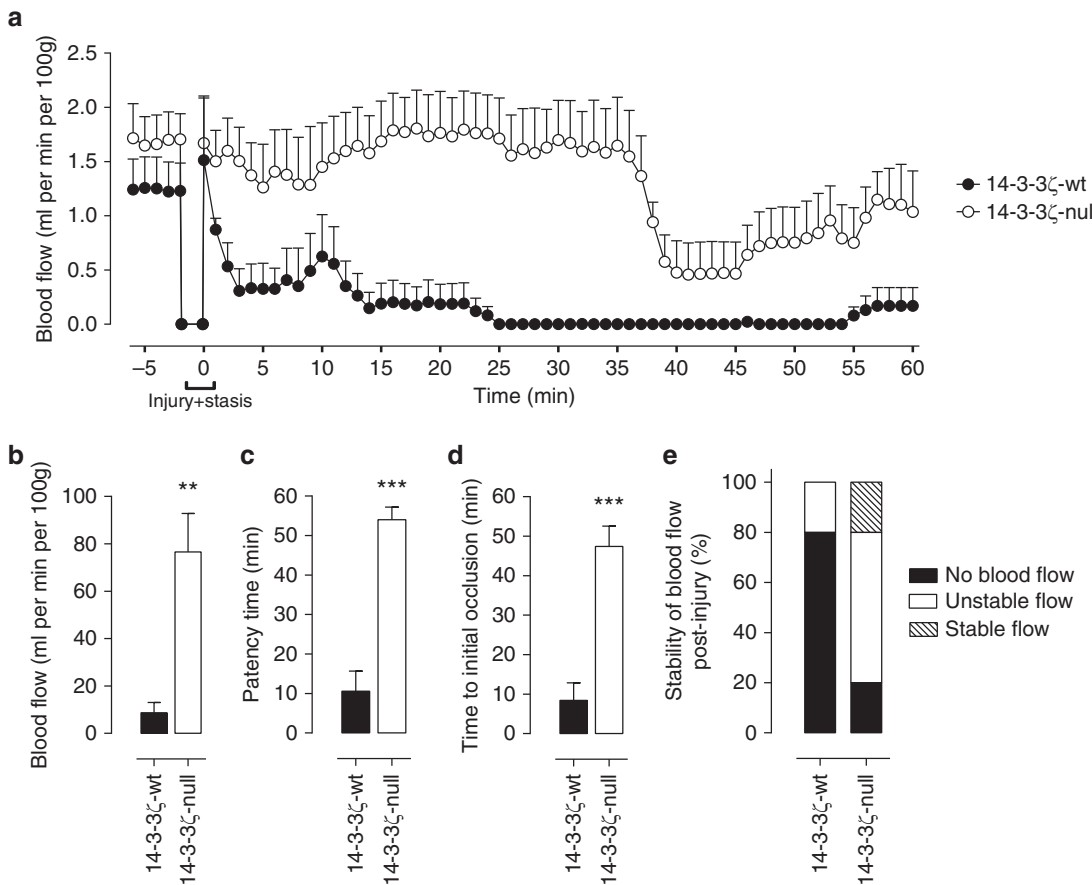

**Figure 1 | 14-3-3ζ deficiency reveals defective arterial thrombosis after electrolytic injury.** (a–d) Electrolytic injury was induced in the carotid artery of 14-3-3ζ-wt (black symbols/bars) or 14-3-3ζ-deficient (14-3-3ζ-null, white symbols/bars) mice, as described under 'Methods'. Carotid artery blood flow was monitored for 60 min following arterial injury (injury + stasis) (a), and further analysed offline. Total blood flow (b), time of vessel patency (c) and time to initial occlusion (d) were quantified over a 60 min period. Note: where vessels did not occlude, vessel patency was recorded as 60 min. Results reflect mean ± s.e.m. (n = 5), with results analysed using an unpaired student t-test (**P < 0.01, ***P < 0.001). (e) The effect of 14-3-3ζ deficiency on stability of blood flow post injury was classified into three categories, as described under 'Methods'.

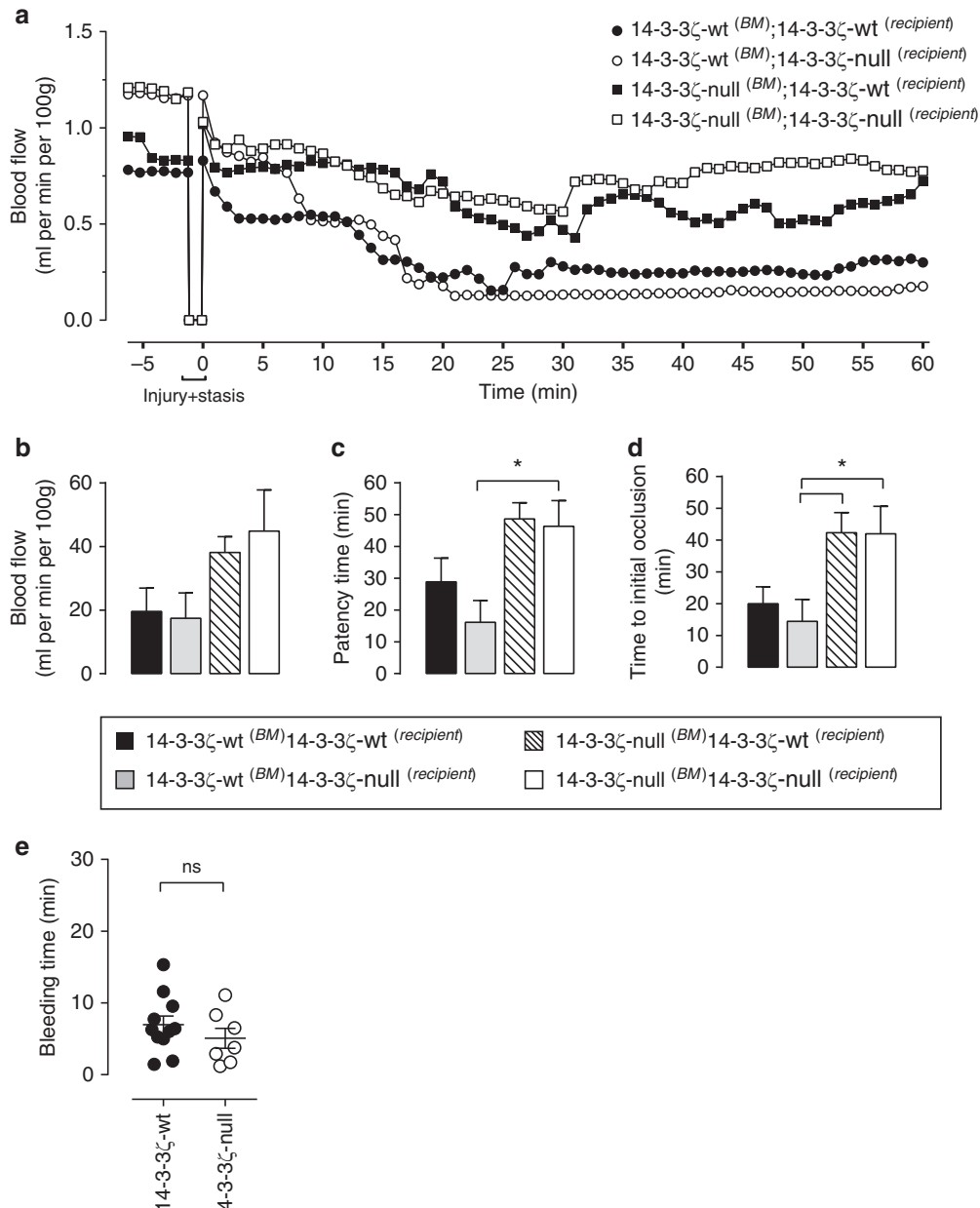

**Figure 2 | Hematopoietic-specific 14-3-3ζ deficiency results in defective arterial thrombosis. (a–d)** Electrolytic injury was induced in the carotid artery of four cohorts of bone marrow-transplanted mice (refer to key): *14-3-3ζ-wt* donor bone marrow (BM) into 14-3-3ζ-wt ($n = 9$) or 14-3-3ζ-null ($n = 8$) recipient mice and *14-3-3ζ-null* donor BM into either 14-3-3ζ-wt ($n = 9$) or 14-3-3ζ-null ($n = 7$) recipients, as described in 'Methods'. Carotid artery blood flow over 60 min (**a**), as well as total blood flow (**b**), time of vessel patency (**c**) and time to initial occlusion (**d**) were quantified over 60 min, as described in the legend to Fig. 1. Results represent mean ± s.e.m. (*$P < 0.05$), where results were analysed using a one-way ANOVA with Sidaks *post hoc* testing. (**e**) Haemostasis was examined using a 3 mm tail lop model of bleeding time, as described under 'Methods'. Results are expressed as time taken to bleeding cessation, and represent the mean ± s.e.m. (14-3-3ζ-wt, $n = 11$; 14-3-3ζ-null, $n = 7$). Results were analysed using an unpaired Student's *t*-test ($^{NS}P > 0.05$).

externalization, platelet-dependent fibrin clot formation was also reduced in both human platelets treated with RB-011 (Fig. 4d), as well as platelets isolated from 14-3-3ζ-deficient mice (Fig. 4e). In control studies, we confirmed that the effects of RB-011 were selective for platelet procoagulant function, as agonist-induced α-granule release (P-selectin expression) and integrin $α_{IIb}β_3$ activation were normal in RB-011-treated platelets (Supplementary Fig. 3a,b). Interestingly, RB-011 had minimal effects on platelet PS exposure (Fig. 4c) and fibrin clot formation in 14-3-3ζ-deficient platelets (Fig. 4f), indicating that 14-3-3ζ is likely to be the main 14-3-3 isoform regulating platelet PS exposure.

PS exposure also occurs during platelet apoptosis, via an activation-independent event linked to the Bcl-2 protein family[26]. However, PS surface expression was normal in 14-3-3ζ deficient platelets treated with the pro-apoptotic BH3 mimetic, ABT-737 (Supplementary Fig. 4a). Moreover, PS exposure in platelets treated with the calcium ionophore A23187 (Supplementary Fig. 4b), which bypasses receptor signalling mechanisms linked to calcium flux, was normal in 14-3-3ζ-wt and 14-3-3ζ-deficient platelets. These findings indicate that the role of 14-3-3ζ is specific to PS exposure induced by potent physiological platelet agonists.

**14-3-3ζ-deficient mice have reduced thrombin generation *in vivo*.** To investigate whether the alterations in platelet PS exposure and procoagulant function *in vitro* reduced thrombin generation *in vivo*, we performed studies in a collagen/epinephrine pulmonary embolism model, a thrombosis model that is both platelet- and coagulation-dependent[27,28]. Mice were intravenously injected with collagen/epinephrine ($1.0\,\text{mg}\,\text{kg}^{-1}$; $60\,\mu\text{g}\,\text{kg}^{-1}$) or an equivalent volume of sterile saline, and blood samples were taken to quantify thrombin generation. Seventy-five per cent of wt mice died within 30 min of collagen/epinephrine

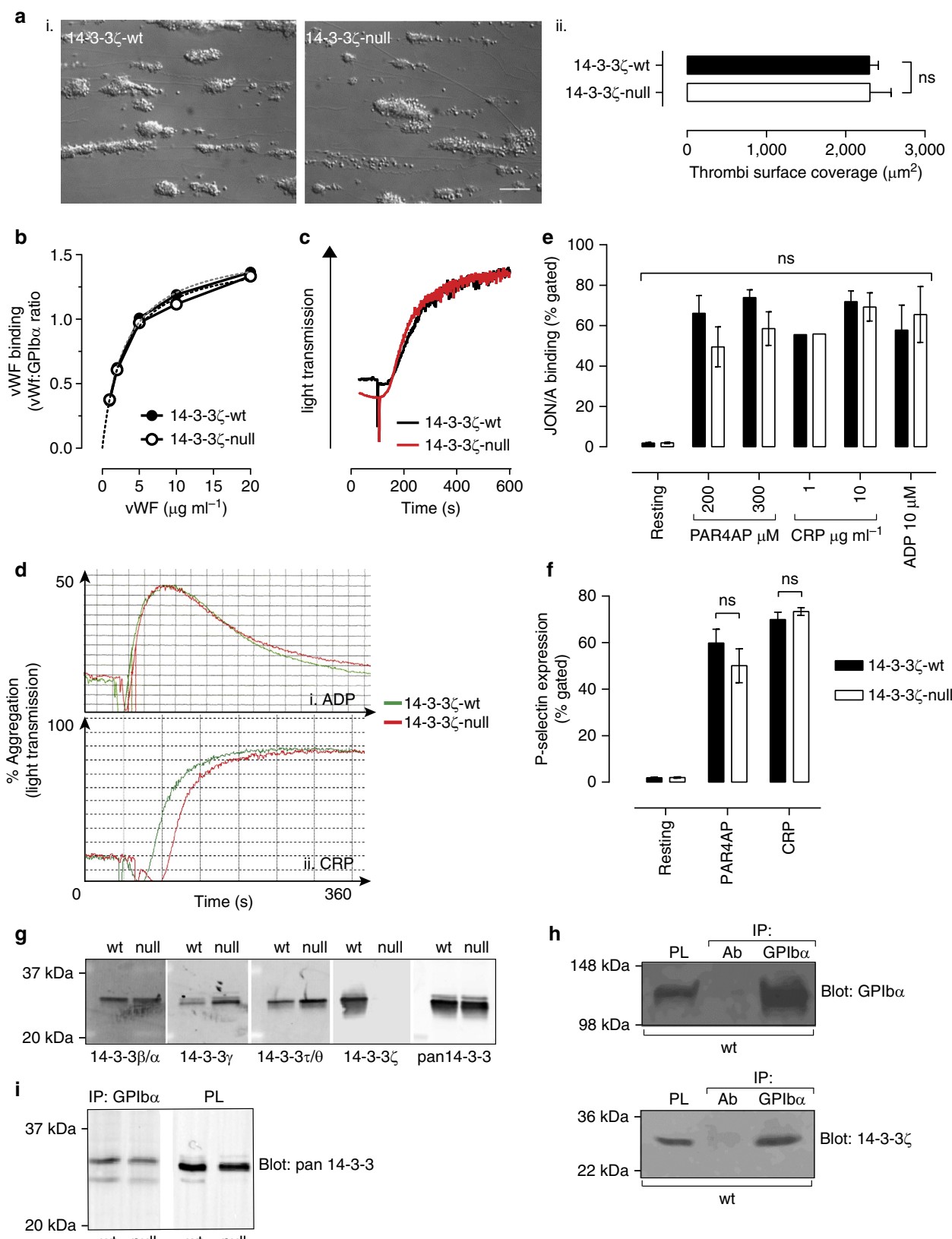

injection, compared with 12.5% of 14-3-3ζ-deficient mice (Fig. 5a). Histological examination of lung tissue demonstrated that the majority of pulmonary vessels in wt mice were completely occluded, whereas 14-3-3ζ-deficient mice demonstrated a significant reduction in occlusive pulmonary thromboemboli (Fig. 5b) and a marked decrease in thrombus burden (Fig. 5b(ii)). ELISA-based quantification of thrombin generation, by measuring thrombin–antithrombin III complexes (TAT), revealed significantly lower levels of TAT in 14-3-3ζ-deficient mice at both 10 and 30 min post challenge (Fig. 5c), despite a similar drop in platelet count in 14-3-3ζ-deficient mice relative to 14-3-3ζ-wt mice (Fig. 5d). These findings are consistent with the possibility that reduced thrombin generation, rather than impaired platelet activation by collagen and epinephrine, is primarily responsible for reduced thrombus formation in 14-3-3ζ-deficient mice.

**Sustained metabolic ATP levels in 14-3-3ζ-deficient platelets.** Agonist-induced platelet PS exposure and procoagulant function are principally induced by opening of the mitochondrial permeability transition pore, as a consequence of high, sustained levels of cytosolic calcium[29]. Such high levels of sustained calcium are toxic for the cell, leading to a specific form of cell death termed regulated cell necrosis[26,30,31]. To investigate whether there were differences in agonist-stimulated cytosolic calcium response between 14-3-3ζ-wt and 14-3-3ζ-deficient platelets, that might explain the differences in the extent of PS exposure, washed platelets were loaded with the calcium indicator dyes, Oregon-Green BAPTA and Fura-Red, prior to platelet stimulation. These studies revealed no significant difference in cytosolic calcium flux in response to platelet stimulation by a broad range of agonists, including thrombin and CRP (Supplementary Fig. 5).

Next, we considered the possibility that 14-3-3ζ may regulate the metabolic function of platelets, since a major feature of necrotic cell death is bioenergetic failure[30]. Quantitative analysis of metabolic ATP in activated platelets revealed that potent platelet stimulation with thrombin/CRP caused an ∼50% depletion of metabolic ATP after 30 min of stimulation in wt platelets (Fig. 6a). In contrast, 14-3-3ζ-deficient platelets demonstrated sustained level of metabolic ATP, with <10% reduction in ATP 30 min post-stimulation with thrombin/CRP (Fig. 6a). Consistent with this, human platelets pretreated with RB-011 also demonstrated more sustained levels of metabolic ATP following agonist challenge (Supplementary Fig. 6a). The phosphorylation status of AMP kinase (AMPK) is a reliable

indicator of metabolic stress, with increased phosphorylation coinciding with ATP depletion[32]. Consistent with the possibility of reduced metabolic stress in 14-3-3ζ-deficient platelets, AMPK phosphorylation was reduced in 14-3-3ζ-deficient platelets relative to 14-3-3ζ-wt controls, following thrombin/CRP stimulation (Fig. 6b,c). Taken together, these findings suggest an important role for 14-3-3ζ in regulating the metabolic function of platelets.

**Elevated mitochondrial respiratory reserve in 14-3-3ζ-deficient platelets.** The preservation of cellular ATP is achieved principally through mitochondrial and glycolytic pathways[33,34]. 14-3-3ζ is known to regulate the enzyme activity of two rate-limiting glycolytic enzymes, 6-phosphofructokinase (6-PFK) and pyruvate kinase (PK). We examined the glycolytic capacity of 14-3-3ζ-wt and 14-3-3ζ-deficient platelets by measuring proton flux (extracellular acidification rate, ECAR) in the presence of glutamine. Using an XFp extracellular flux analyser (Seahorse Bioscience), the glycolytic rate was determined by quantifying ECAR in resting platelets, before and after exposure to glucose (10 mM). Maximum glycolytic capacity was determined by treating platelets with the $F_0F_1$ ATP synthase inhibitor Oligomycin A, and the contribution of the glycolytic pathway was confirmed by inhibiting glycolysis with 2-deoxyglucose (2-DG). In both resting and thrombin/CRP-stimulated platelets, ECAR measurements were identical in 14-3-3ζ-wt and 14-3-3ζ-deficient platelets under all experimental conditions tested (Supplementary Fig. 7a,b), indicating that dysregulated glycolysis was not a prominent feature of 14-3-3ζ-deficient platelets. Consistent with this, measurement of 6-PFK and PK activity in lysates generated from resting and stimulated platelets revealed no differences in the kinetics of both enzymes (Supplementary Fig. 7c,d), suggesting that the alterations in metabolic ATP in 14-3-3ζ-deficient platelets were not primarily due to alterations in glycolysis.

14-3-3ζ has been demonstrated to regulate mitochondrial ATP generation in plant cells, by modifying the activity of mitochondrial complex V ($F_0F_1$ ATP synthase)[35,36]. To examine a potential contribution of mitochondria in the observed 14-3-3ζ-deficient phenotype, platelets from 14-3-3ζ-wt or 14-3-3ζ-deficient mice were incubated with the $F_0F_1$ ATP synthase inhibitor, oligomycin A. As demonstrated in Fig. 7, pre-treatment with oligomycin ablated the difference in agonist-stimulated PS exposure between wt and 14-3-3ζ-deficient platelets (Fig. 7a). Consistent with this, oligomycin-pre-

**Figure 3 | GPIbα and platelet function are unperturbed by 14-3-3ζ deficiency.** (**a**) Hirudin-anticoagulated blood from 14-3-3ζ-wt and 14-3-3ζ-deficient (14-3-3ζ-null) mice was perfused through collagen-coated (250 μg ml$^{-1}$ Type I) microslides at 1,800 s$^{-1}$ for 5 min. (i) DIC images are taken from 1 representative of four independent experiments (scale bar, 20 μm). (ii) Thrombi surface coverage was quantified using ImageJ software (4–5 fields analysed per flow). The histogram depicts the mean ± s.e.m. (n = 5; analysed using two-way ANOVA with Bonferroni's *post hoc* testing). (**b**) VWF binding was measured as described under 'Methods'. The graph represents one of three independent experiments. (**c**) Aggregation of washed mouse platelets in the presence of human VWF (10 μg ml$^{-1}$) and botrocetin (10 μg ml$^{-1}$), with stirring. The graph depicts aggregation traces from one representative of three independent experiments. (**d**) Comparative aggregation of washed 14-3-3ζ-wt and 14-3-3ζ-null platelets in response to CRP or ADP ((i) ADP 1 μM; (ii) CRP 20 ng ml$^{-1}$). Results are taken from one representative experiment. A histogram depicting the mean ± s.e.m. (n = 3; analysed using a two-way ANOVA with Bonferroni's *post hoc* testing), is presented in Supplementary Fig. 2b. (**e,f**) Diluted whole blood samples from 14-3-3ζ-wt (black bars) or 14-3-3ζ-null (white bars) mice were incubated with PE-JON/A (**e**) or FITC-anti P-selectin antibody (**f**), as described under 'Methods', to examine integrin α$_{IIb}$β$_3$ and degranulation of platelet α-granules, respectively, and analysed by flow cytometry, following incubation with the indicated agonist/concentration for 15 min. Results depict the % of gated platelets positive for antibody binding and are expressed as the mean ± s.e.m. ((**e**) PAR4P: n = 10; CRP: n = 3; ADP: n = 4; (**f**) PAR4P: n = 10; CRP: n = 5), where $^{NS}P > 0.05$. Note: no significant difference was observed in the geometric mean of fluorescence intensity for the same experiments. (**g–i**) Resting whole cell lysates were prepared from 14-3-3ζ-wt (wt) and 14-3-3ζ-null (null) washed platelets. (**g**) Expression levels of 14-3-3 isoforms were compared using SDS–PAGE and immunoblot analysis, as described under 'Methods', with immunoblots probed with the indicated 14-3-3 isoform-selective or pan-14-3-3 antibodies. These studies confirm deletion of 14-3-3ζ protein in the 14-3-3ζ-null mice, with some upregulation of 14-3-3γ. (**h,i**) Association of 14-3-3 proteins with the GPIbα cytoplasmic tail in the absence of 14-3-3ζ—GPIbα was immunoprecipitated from 14-3-3ζ-wt and 14-3-3ζ-null platelet lysates as described under 'Methods', and analysed via immunoblotting, using (**h**) anti-14-3-3ζ or (**i**) anti-pan-14-3-3. Immunoblots are taken from one representative of three independent experiments.

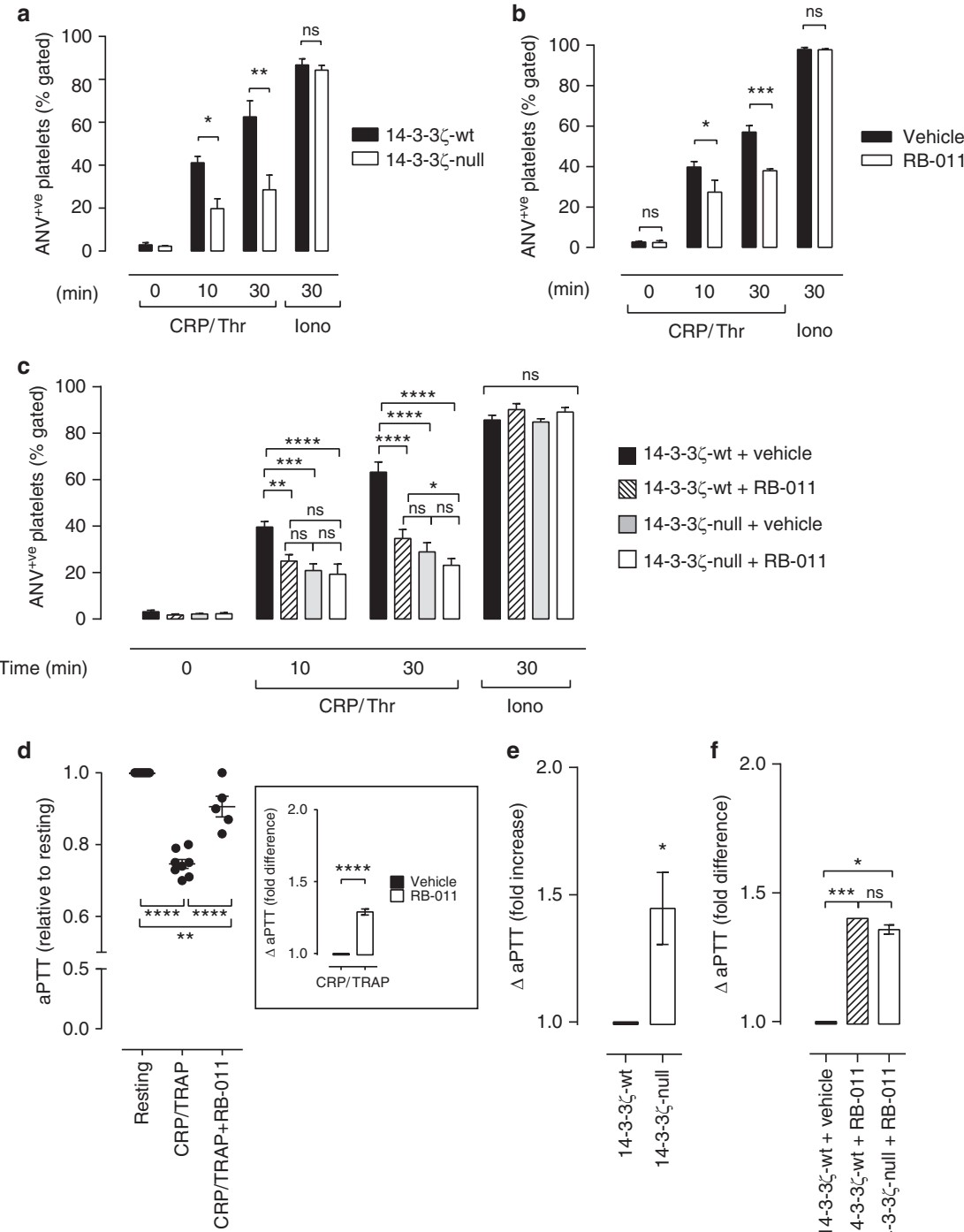

**Figure 4 | Reduced PS exposure and thrombin generation in 14-3-3ζ-deficient platelets.** (**a**–**c**) PS exposure was quantified in anticoagulated whole blood isolated from 14-3-3ζ-wt or 14-3-3ζ-null mice (**a**), or in washed platelets isolated from either 14-3-3ζ-wt and 14-3-3ζ-null mice (**c**) or healthy human donors (**b**). Where indicated, washed platelets were treated with a 14-3-3 dimer destabilizer (RB-011, 10 μM) or vehicle (sodium mesylate salt), prior to activation. In whole blood, PS exposure was quantified following stimulation of platelets with buffer (resting), CRP/Thrombin (CRP (0.25 μg ml$^{-1}$)/Thr (0.5 U ml$^{-1}$)) or calcium ionophore A23187 (Iono, 1 μm), for the indicated times. Washed platelets were stimulated with CRP (0.25 μg ml$^{-1}$)/TRAP (1 μM) or CRP (0.25 μg ml$^{-1}$)/PAR4 activating peptide (200 μM) (for human and mouse platelets, respectively) for the indicated times. PS was detected through measurement of Alexa-488-labelled Annexin V binding (ANV$^{+ve}$), as described under 'Methods'. Results are expressed as the mean ± s.e.m. ($n=8$), where ****$P<0.0001$; ***$P<0.001$; **$P<0.01$; *$P<0.05$; $^{NS}P>0.05$. (**d**–**f**) Procoagulant function of washed human (**d**) or mouse platelets (**e**,**f**) was examined using a modified aPTT assay with Ellagic acid as a soluble activator, as described under 'Methods'. Resting platelets or those activated for 10 min with CRP/TRAP (**d**) or CRP/PAR4-activating peptide (**e**,**f**) were added as a source of phospholipid. Results depict the mean ± s.e.m. ($n=4$; where ****$P<0.0001$; ***$P<0.001$; **$P<0.01$; *$P<0.05$; $^{NS}P>0.05$). Results represent the aPTT in vehicle- and RB-011-treated activated platelets, relative to resting platelets (**d**), the fold difference in aPTT when comparing activated vehicle- and RB-011-treated human (**d**, inset) or 14-3-3ζ-wt and 14-3-3ζ-null mouse platelets (**f**), or the fold difference in aPTT when comparing activated 14-3-3ζ-wt and 14-3-3ζ-null mouse platelets (**e**). Where appropriate, results were analysed using either unpaired Student's *t*-test, one-way or two-way ANOVA (with Tukey's or Bonferroni's *post hoc* testing).

treatment also negated the sustained metabolic ATP levels observed in 14-3-3ζ-deficient platelets (Fig. 6a). Moreover, while 14-3-3ζ-deficient platelets were able to maintain mitochondrial membrane potential ($\Delta\psi_m$) following potent stimulation when compared with their wt counterparts, oligomycin pre-treatment also rendered both platelet genotypes equally sensitive to mitochondrial membrane depolarization (decreased $\Delta\psi_m$) (Fig. 7b). These studies demonstrate that inhibition of mitochondrial function through direct blockade of the $F_0F_1$ ATP synthase negates the impact of 14-3-3ζ deficiency on platelet bioenergetics, PS exposure and $\Delta\psi_m$, indicating that altered mitochondrial metabolism contributes to the platelet phenotype.

To examine directly mitochondria bioenergetics in 14-3-3ζ-deficient platelets, we employed the XFp extracellular flux analyser, which allows determination of mitochondrial function through the measurement of oxygen consumption rate (OCR), in the extracellular milieu of cells. As demonstrated in Fig. 8a–d, basal OCR in platelets isolated from both genotypes were similar. We further profiled mitochondrial function in both genotypes by examining OCR following the addition of well-defined inhibitors of the electron transport chain, including oligomycin, FCCP (uncoupler) and Rotenone/Antimycin A (Complex I/III). Resting platelets isolated from either 14-3-3ζ-wt or 14-3-3ζ-deficient mice demonstrated a standard metabolic response to each inhibitor, with a reduction in OCR on inhibition of $F_0F_1$ ATP synthase (ATP-linked respiration), a rapid increase in OCR on FCCP-mediated uncoupling of the electron transport chain (maximal respiration), and finally a complete inhibition of OCR by inhibiting complexes I and III (Fig. 8a,b). As with basal OCR, the rate and extent of OCR in response to all treatments was similar in resting platelets from both genotypes (Fig. 8a,b). To test

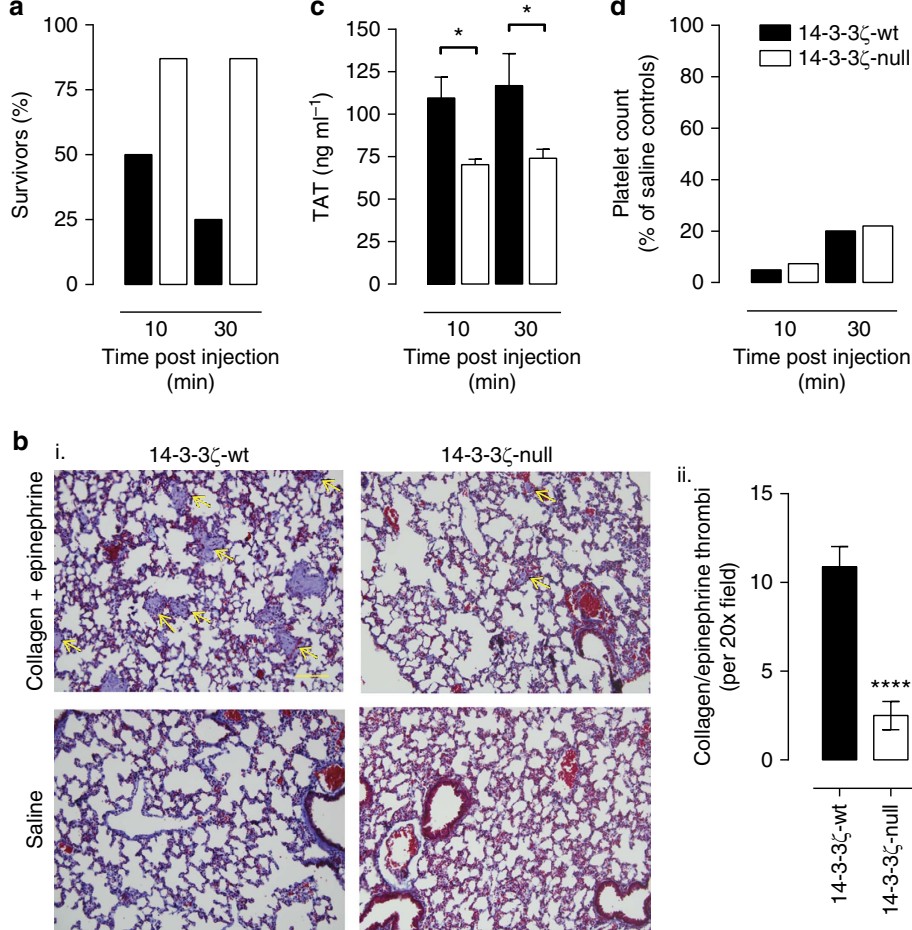

**Figure 5 | 14-3-3ζ-deficient mice are protected from pulmonary thromboembolism.** Thromboembolism was induced in 14-3-3ζ-wt and 14-3-3ζ-deficient (14-3-3ζ-null) mice by intravenous injection of collagen and epinephrine, as described under 'Methods'. In control studies, mice were injected with saline (Control). (**a**) Number (percentage, %) of mice surviving at 10 and 30 min post collagen/epinephrine injection, n = 8. (**b**(i)) Lungs were collected, fixed in 10% formalin and 5 mm sections cut from paraffin-embedded lung tissue. Sections were stained with Masson Trichrome and analysed for the presence of thrombi on an Olympus BX51 microscope; images were captured using a DP70 camera and DP70-BSW software (scale bar = 100 μm). 14-3-3ζ-wt mice had extensive occlusive platelet thrombi (yellow arrows), which were significantly reduced/absent in surviving 14-3-3ζ-null mice and absent in all saline controls. Note: 14-3-3ζ-wt lung histology derived from a mouse with platelet count = 0 × 10³ μl at time of death 9′00″; 14-3-3ζ-null lung histology derived from a surviving mouse with platelet count = 90 × 10³ μl, taken at 30′00″; (ii) clot burden in the lungs of both 14-3-3ζ-wt and 14-3-3ζ-null mice was assessed by quantification of the number of occluded vessels per field of view. Saline control images were found to have no occluded vessels. These results represent the mean ± s.e.m. (n = 3–4 mice, 3–5 sections imaged per mouse). Results were analysed using an unpaired Student's t-test, where ****P < 0.0001. (**c**) In vivo thrombin generation was measured in plasma samples taken at 10 and 30 min post injection using a thrombin–antithrombin (TAT) ELISA; n = 4–8 since some mice had already died at early time points post injection. Results were analysed using one-way ANOVA (Bonferroni's post hoc), *P < 0.05. (**d**) Blood samples were taken from mice at 10 and 30 min post injection and platelet counts were compared with saline controls at the same time points.

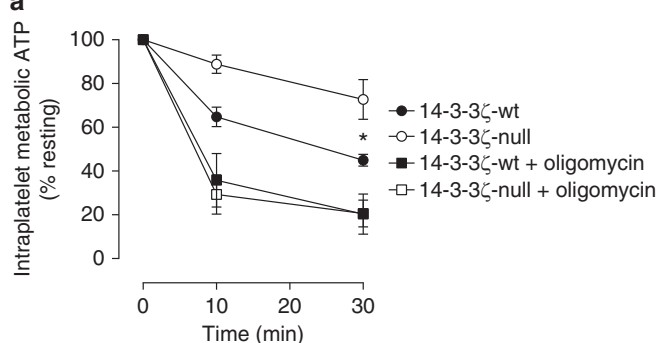

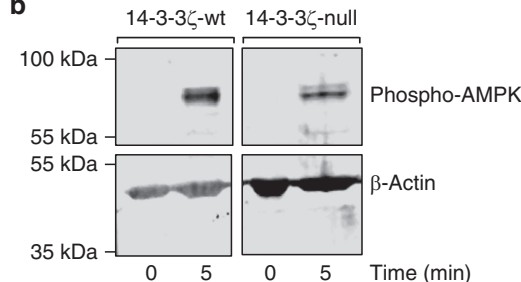

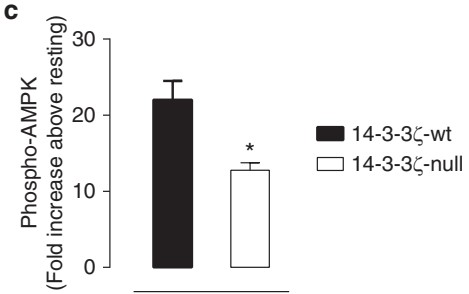

**Figure 6 | 14-3-3ζ-deficient platelets demonstrate sustained metabolic ATP following potent activation.** (**a**) Washed platelets ($1 \times 10^8 \, \text{ml}^{-1}$) were isolated from 14-3-3ζ-wt (black symbols) or 14-3-3ζ-deficient (14-3-3ζ-null) (white symbols). Where indicated (squares), platelets were pre-incubated with oligomycin ($1 \, \mu\text{M}$), prior to stimulation. Metabolic ATP was extracted and quantified in unstimulated (Resting) and stimulated platelets with a mix of CRP ($0.25 \, \mu\text{g} \, \text{ml}^{-1}$) and thrombin ($0.5 \, \text{U} \, \text{ml}^{-1}$) for up to 1,800 s as described in 'Methods' section. Results are expressed as the percentage (%) change in ATP compared with resting control levels over the indicated times, and represent the mean ± s.e.m. ($n = 4$). A two-way ANOVA was used to determine statistical difference between 14-3-3ζ-wt and 14-3-3ζ-null platelets ($^{NS}P > 0.05$; $^*P < 0.05$). (**b,c**) Western blot analysis of phospho-AMPK in 14-3-3ζ-wt and 14-3-3ζ-null platelets: Washed platelets ($3 \times 10^8 \, \text{ml}^{-1}$) were isolated from 14-3-3ζ-wt and 14-3-3ζ-null mice and activated with CRP/thrombin ($0.25 \, \mu\text{g} \, \text{ml}^{-1}$, $1 \, \text{U} \, \text{ml}^{-1}$). After activation, platelets were lysed using RIPA buffer. Platelet lysates were analysed by western blot analysis of phospho-AMPK protein using a phosphospecific antibody (40H9). The western blot in **b** is taken from one representative of three independent experiments. Total β-actin was used as a loading control, and quantification of phospho-AMPK signal was expressed as a ratio of protein loading (**c**, mean ± s.e.m., $n = 3$), using LICOR technology. A Student's $t$-test was used to determine statistical difference between 14-3-3ζ-wt and 14-3-3ζ-null platelets ($^*P < 0.05$). Where appropriate, results were analysed using either unpaired Student's $t$-test, two-way ANOVA with Tukey's *post hoc* testing.

mitochondrial function in platelets stimulated with potent platelet agonists, identical measurements were obtained from 14-3-3ζ-wt and 14-3-3ζ-deficient platelets stimulated with a combination

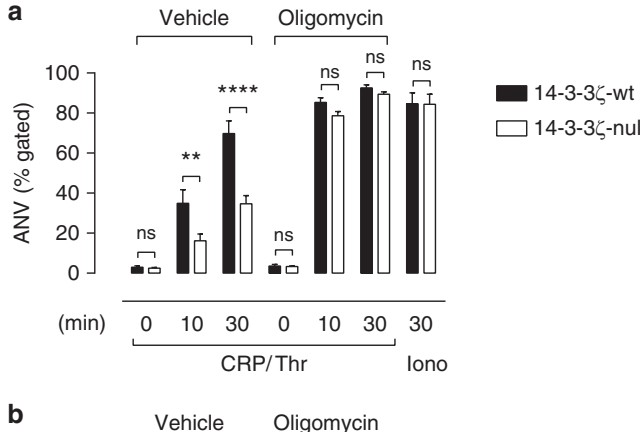

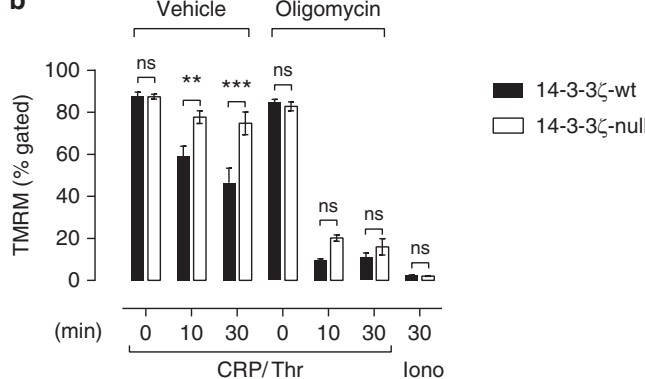

**Figure 7 | Oligomycin negates the reduction in PS exposure in 14-3-3ζ-deficient platelets.** Anticoagulated whole blood was isolated from 14-3-3ζ-wt (black bars) or 14-3-3ζ-deficient (14-3-3ζ-null, white bars) mice. Exposure of PS was quantified through measurement of Alexa-488-labelled Annexin V (ANV) (**a**) and mitochondrial membrane potential measured by TMRM staining (**b**), as described under 'Methods'. PS and mitochondrial membrane potential were quantified in the presence of buffer alone (vehicle) or oligomycin A ($1 \, \mu\text{M}$) (oligomycin), following stimulation of platelets with CRP/thrombin (CRP ($0.25 \, \mu\text{g} \, \text{ml}^{-1}$)/Thr ($0.5 \, \text{U} \, \text{ml}^{-1}$)) or calcium ionophore A23187 (Iono, $1 \, \mu\text{m}$), for the indicated times. Results are expressed as the mean ± s.e.m. ($n = 8$), and analysed using two-way ANOVA with Bonferroni's *post hoc* testing where $^{****}P < 0.0001$; $^{***}P < 0.005$; $^{**}P < 0.01$; $^{NS}P > 0.05$.

of CRP ($0.25 \, \mu\text{g} \, \text{ml}^{-1}$) and thrombin ($0.5 \, \text{U} \, \text{ml}^{-1}$). Agonist stimulation of platelets resulted in an increased ATP-linked respiration, which was enhanced in 14-3-3ζ-deficient platelets, along with a 1.7-fold increase in respiratory reserve capacity (Fig. 8a,c,d). Similar findings were demonstrated in human platelets pretreated with the 14-3-3 dimer destabilizer RB-011 (Supplementary Fig. 6b). These findings support the notion that 14-3-3ζ-deficient platelets have an increased capacity for mitochondrial ATP generation during periods of metabolic stress induced by potent platelet agonists.

## Discussion

Platelets play a key role in propagating coagulation reactions within a developing thrombus, necessary for efficient thrombus growth and stability[5]. This is primarily dependent on their ability to express negatively charged phospholipids on their plasma membrane, particularly PS, through mitochondrial-driven cell death pathways linked to the activation of phospholipid scramblases[26,30]. In this manuscript, we describe an important role for the 14-3-3ζ adaptor protein in regulating platelet PS exposure and procoagulant function following potent platelet activation. This non-redundant role for 14-3-3ζ is linked to

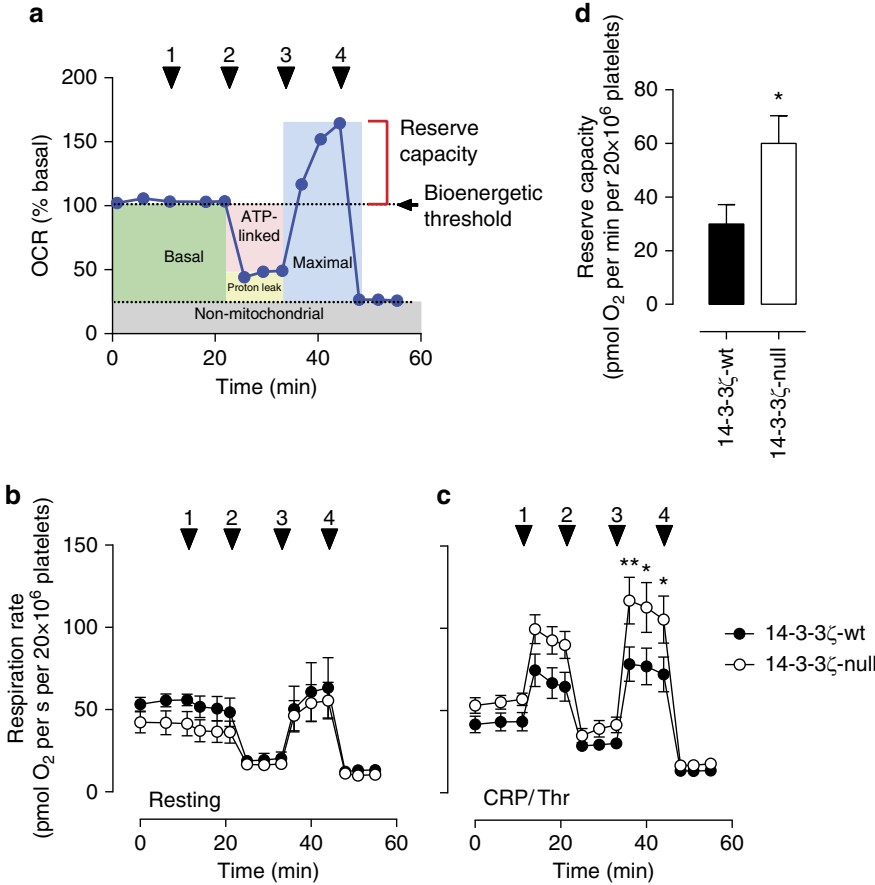

**Figure 8 | 14-3-3ζ deficiency enhances platelet mitochondrial respiratory reserve capacity.** Washed platelets were isolated from 14-3-3ζ-wt (black circles/bars) or 14-3-3ζ-deficient (14-3-3ζ-null, white circles/bars) mice. Mitochondrial respiration was measured in DMEM-modified media using the Seahorse XFp analyzer, according to the manufacturer's instructions. (**a**) Representative trace identifying the typical pattern of OCR, depicting basal level, proton leak, ATP production, maximal respiration and reserve capacity, following inhibition of individual electron transport chain complexes (indicated by arrowheads), including: (1) vehicle or agonist; (2) oligomycin; (3) FCCP; and (4) rotenone/antimycin A. Platelets were assayed for oxygen consumption in unstimulated (**b**) or CRP/Thr-stimulated ($0.25\,\mu g\,ml^{-1}/0.5\,U\,ml^{-1}$) (**c,d**) conditions. (**d**) Reserve capacity in CRP/Thr-stimulated platelets was calculated based on the difference between the basal and maximal OCR (**c**). Results represent the mean ± s.e.m. ($n = 8$), and were analysed using two-way ANOVA (Bonferroni's *post hoc*), where **$P < 0.01$; *$P < 0.05$.

altered mitochondrial bioenergetics, with deficiency of 14-3-3ζ leading to an enhanced capacity to respond to metabolic stress as a consequence of increased mitochondrial respiratory reserve. These findings highlight a new function for 14-3-3 adaptor proteins in regulating mitochondrial bioenergetics linked to PS exposure, independent of apoptosis. They also raise the interesting possibility that therapeutic targeting of platelet mitochondrial metabolism may represent a new approach to reduce thrombosis.

Most of the available evidence supporting an important signalling role for 14-3-3ζ in platelets relates to GPIbα (refs 10–13,16,18,37–42). This conclusion is partly based on mutagenesis strategies that have targeted the 14-3-3ζ binding site on the C-terminal domain of GPIbα (refs 11–13,16,39,41,42). CHO cells transfected with GPIb mutants that are incapable of binding 14-3-3ζ have revealed reduced GPIb-dependent integrin $\alpha_{IIb}\beta_3$ activation[11,12]. Furthermore, studies using a membrane-permeable phosphorylated peptide inhibitor of the 14-3-3ζ–GPIbα interaction (MPαC) have demonstrated reduced VWF binding to GPIbα and decreased platelet adhesion under flow conditions[13]. The interpretation of these studies with respect to 14-3-3ζ signalling is complicated by the possibility that these approaches may have removed the interaction of all 14-3-3 isoforms with GPIbα. Mangin *et al.*[18] demonstrated that all

isoforms of 14-3-3 present in platelets have the capacity to bind GPIbα through the same C-terminal GPIb binding domains utilized by 14-3-3ζ (residues 580–590 and 606–610). It is likely that the remaining 14-3-3 isoforms present in platelets are able to compensate for 14-3-3ζ deficiency and promote GPIb-dependent integrin $\alpha_{IIb}\beta_3$ activation.

Much speculation has surrounded the signalling mechanisms leading to the development of platelet procoagulant function, with a growing body of evidence supporting a central role for mitochondrial-driven cell death pathways in this process[30]. Two cell death pathways can induce platelet PS exposure and procoagulant function—programmed cell apoptosis and regulated cell necrosis[26]. The former pathway plays a central role in regulating the lifespan of circulating quiescent platelets[43], which we show here is unaffected in 14-3-3ζ-deficient platelets. Interestingly, our own studies using mouse platelets deficient in the proapoptotic Bcl-2 family proteins Bak and Bax have revealed that development of activation-dependent PS exposure is unperturbed, instead demonstrating that necrotic-like cell death pathways are principally involved in regulating agonist-induced development of procoagulant platelets[26]. Cell necrosis is typically caused by bioenergetic failure of the cell and the studies presented in this manuscript are consistent with this concept, demonstrating that reduced platelet PS exposure in

14-3-3ζ-deficient platelets correlates with enhanced mitochondrial bioenergetic function, sustained levels of metabolic ATP and reduced AMPK activation. To our knowledge there are no reports of 14-3-3 adaptor proteins influencing regulated cell necrosis pathways linked to bioenergetic function and PS exposure in any cell type.

Mitochondrial ATP production is important to meet the increased energy demands of activated platelets[33]. Direct evidence that mitochondrial bioenergetics were dysregulated in 14-3-3ζ-deficient platelets was based on our extracellular flux analysis, which identified increased mitochondrial respiratory reserve capacity in these cells, under activation conditions. This respiratory reserve capacity is typically required during periods of metabolic or oxidative stress, as may occur following potent platelet stimulation. Interestingly, the respiratory reserve capacity of platelets is relatively low compared with other hemopoietic cells, representing only 20–30% of total mitochondrial respiratory function[44]. It is notable that the regulatory systems controlling intracellular free calcium levels consume a major portion of the cell's energy supply (up to 90%). Thus it is not surprising that high, sustained calcium levels induced by strong platelet agonists would be expected to lead to ATP depletion and an eventual breach of the respiratory reserve, promoting cell death. The factors regulating the volume of the reserve capacity and its biological relevance are not well understood. Our findings demonstrate for the first time the involvement of 14-3-3ζ in regulating the mitochondrial reserve capacity, and suggest an important biological function in the context of platelet procoagulant function and thrombosis.

Our findings that oligomycin A can completely eliminate the impact of 14-3-3ζ deficiency on platelet bioenergetics suggest that 14-3-3ζ has a major non-redundant role in regulating mitochondrial ATP production. In this context, it is interesting to note that the target of oligomycin, the $F_0F_1$ ATP synthase has been recently implicated as an integral component of the mitochondrial membrane permeability transition pore (mPTP)[45–48]. This is a key structure facilitating permeability transition and functional decline of mitochondria under conditions of cellular stress, including elevated calcium, ROS and ATP depletion. Studies in barley sprouts have identified a direct interaction between 14-3-3 proteins and the β-subunit of the $F_1$ domain of the $F_0F_1$ ATP synthase, resulting in downregulation of enzyme activity, decreasing the generation of ATP[49,50]. It is tempting to speculate that the platelet phenotype observed in 14-3-3ζ-deficient mice may be due in part to a direct interaction between 14-3-3ζ with the $F_0F_1$ ATP synthase in mammalian cells. However, the demonstration of enhanced respiratory reserve capacity in 14-3-3ζ-deficient platelets occurs in the presence of oligomycin (a direct inhibitor of the $F_0F_1$ ATP synthase), suggests that additional 14-3-3 targets may be involved in regulating mitochondrial bioenergetics.

The demonstration that 14-3-3ζ deficiency in platelets is not associated with spontaneous or surgical bleeding, yet protects mice from arterial thrombi, raises the interesting possibility that therapeutic targeting of 14-3-3ζ, may represent a safe and effective antithrombotic strategy. A growing body of evidence supports the concept that reducing the procoagulant function of platelets may help reduce propagation of arterial thrombi, without impacting on haemostasis[51,52]. 14-3-3ζ is ubiquitously expressed and shares a high degree of homology with several other 14-3-3 isoforms, so the ability to develop isoform-selective inhibitors with limited toxicity remains a challenge. Nonetheless, there is growing interest in developing 14-3-3ζ inhibitors for the treatment of cancer[53,54], since overexpression of 14-3-3ζ occurs commonly in cancer[53,54], and is associated with disease recurrence[53–55] and resistance to chemotherapy[56–58]. Many of the cancers associated with dysregulated 14-3-3ζ signalling[53], such as stomach, lung, pancreatic, breast, diffuse large B-cell lymphoma, ovarian and lung have an increased risk of venous thromboembolism[59–61], with fatal pulmonary embolism representing a significant cause of premature death in these patients[62]. The demonstration that 14-3-3ζ deficiency protects from fatal pulmonary embolism in mice raises the interesting possibility that 14-3-3 inhibitors may have the added benefit of reducing thrombotic complications during cancer therapy.

## Methods

**Materials.** Antibodies and reagents—rat monoclonal Ab's Xia.B2 (anti-GPIbα), Xia.G5-FITC (anti-GPIbα), Xia.C3-FITC (anti-GPIbβ), Leo.F2-FITC (anti-integrin $\alpha_{IIb}\beta_3$), JAQ.1-FITC (anti-GPVI) and JonA-PE (anti-activated integrin $\alpha_{IIb}\beta_3$) were from Emfret Analytics (Eibelstadt, Germany). Anti-mouse P-selectin-FITC was from Becton Dickinson. OG-FGN, sulfo-NHS biotin, ADP, calcium ionophore A23187, probenecid, annexin V and ellagic acid were from Sigma-Aldrich (St Louis, MO). Anti-Phospho-AMPKα (Thr172) rabbit mAb (40H9), β-Actin (8H10D10) mouse mAb, antibodies selective for anti-14-3-3 β/α(#9636), anti-14-3-3 γ (#9637), anti-14-3-3 τ (#9638) and anti-rabbit IgG HRP-linked antibody (#7074) were from Cell Signaling Technology (USA); anti-14-3-3ζ (C-16, sc-1019) or pan-14-3-3 isoform antibodies (K19, sc-629) were from Santa Cruz Biotechnology, Inc (USA). Annexin V was fluorescently labelled in house using Alexa-488 (Molecular Probes, Eugene, OR). IRDye 680LT Donkey anti-Mouse IgG and IRDye 800CW Donkey anti-Rabbit IgG secondary antibodies were from LI-COR Biotechnology (Lincoln, USA). Tetramethylrhodamine methyl ester (TMRM) was from Molecular Probes (Eugene, OR). Thrombin and the thrombin-anti-thrombin (TAT) ELISA kit were from Dade Behring (Marburg, Germany). PAR-4 agonist peptide AYPGKF was from Auspep (Parkville, Victoria, Australia). U46619 was from Merck (Australia). Collagen-related peptide (CRP) was from Richard Farndale (Oxford, UK). ENLITEN Luciferase/Luciferin reagent was purchased from Promega (Sydney, Australia). Seahorse Bioscience XF reagents were purchased from 3 In Vitro Technologies Pty. Ltd (Queensland, Australia). The 14-3-3 dimer destabilizer RB-011 was synthesized as described previously[23]. Commercially available colorimetric enzyme assay kits to quantify pyruvate kinase (ab83432) and 6-phosphofructokinase (ab155898) activity were purchased from Abcam (Victoria, Australia). All other reagents were from sources described previously[22,26,63]. Working concentrations of all antibodies and reagents used for each application are specified in the following specific method section(s).

**Mouse strains.** 14-3-3ζ-deficient mice were generated using gene trap constructs as described by Cheah et al.[19] Animal experiments were conducted in accordance with the guidelines of SA Pathology and the Central Adelaide Local Health Network (CALHN) and University of Sydney Animal Ethics Committees, according to ethics applications E/0475/2006/M and E/0936/2010/M approved by the Alfred Medical Research & Education Precinct Animal Ethics Committee, and ethics application 2014/620 approved by the University of Sydney Animal Ethics Committee. Platelet counts were performed on a Sysmex KX-21N automated haematology analyser (Roche Diagnostics, Australia) using sub-mandibular blood samples collected into EDTA tubes.

**Generation of 14-3-3ζ bone marrow-transplanted mice.** Sv129 (14-3-3ζ-wt and 14-3-3ζ-null) female mice of at least 8 weeks of age were irradiated with 1,200 cGy using a deep X-ray irradiation machine. Irradiated mice were transplanted with intravenous injections of $15 \times 10^6$ bone marrow cells, collected from tibiae and femurs from male (14-3-3ζ-wt and 14-3-3ζ-deficient) mice, resuspended in phosphate-buffered saline (PBS). Volumes for injections were 0.5 ml per transfusion. The percentage of male marrow in total host marrow cells was determined at 8 weeks post transplant using fluorescent in situ hybridization (FISH) probing for the presence of the Y-chromosome. The FISH probe was prepared as reported by Donnelly and colleagues[64], wherein a Y-chromosome probe was produced using degenerate oligonucleotide primed-PCR (oligonucleotide—CCGACTCGAGNNNNNNATGTGG) with a template derived from murine Y-chromosome.

**Determination of platelet lifespan.** Mice were injected intravenously (tail vein) with 600 μg sulpho-NHS biotin. At 4 h post injection and time points thereafter, blood was collected from the tail vein and the percentage of CD41-positive events that were streptavidin-PE positive was quantitated by flow cytometry[43].

**Flow cytometry.** Anti-coagulated whole blood samples were diluted 1:20 in Tyrode's buffer[65]. For receptor expression analysis, samples were stained with fluorophore-conjugated monoclonal antibodies for 10 min at room temperature and analysed on a FACSCalibur flow cytometer (BD Biosciences).

**VWF binding.** Washed mouse platelets ($2 \times 10^7 \, ml^{-1}$) in Tyrode's buffer[65] supplemented with 1% BSA were incubated with Alexa488-VWF ($1–20 \, \mu g \, ml^{-1}$) in the presence or absence of the VWF-modulator botrocetin ($10 \, \mu g \, ml^{-1}$) for 30 min at room temperature, then diluted to $400 \, \mu l$ and analysed immediately by flow cytometry. Results are expressed as VWF:GPIbα binding ratio using Xia.G5-FITC mAb ($5 \, \mu l$ mAb stock solution in $25 \, \mu l$ total assay volume), according to the manufacturer's instructions, to measure GPIbα expression levels on 14-3-3ζ-wt and 14-3-3ζ-deficient platelets.

**P-selectin expression and integrin $\alpha_{IIb}\beta_3$ activation.** Diluted mouse blood samples were pre-incubated with either rat anti-mouse P-selectin-FITC Ab ($1 \, \mu g \, ml^{-1}$), or JON/A-PE ($5 \, \mu l$ stock solution in $25 \, \mu l$ total assay volume) according to the manufacturer's instructions. Washed human platelets ($1 \times 10^8 \, ml^{-1}$) were incubated with anti-PAC-1-FITC monoclonal antibody ($0.25 \, \mu g \, ml^{-1}$, BD biosciences). Samples were stimulated with various agonists for the indicated times at $37 \, °C$, diluted with Tyrode's buffer and analysed by flow cytometry.

**Measurement of Annexin V binding.** Washed mouse platelets ($5.0 \times 10^7 \, ml^{-1}$) were incubated with CRP or thrombin (either alone or in combination (CRP/Thr), at the indicated doses, calcium ionophore A23187 ($0.25–1.0 \, \mu M$) or ABT-737 ($1 \, \mu M$) for 30–60 min (where indicated) in the presence of Alexa488-Annexin V ($1–2 \, \mu g \, ml^{-1}$). Samples were diluted into Tyrode's buffer and analysed immediately by flow cytometry.

**Measurement of mitochondrial membrane potential.** Washed mouse platelets ($1 \times 10^8 \, ml^{-1}$) in PWB were incubated with TMRM ($0.5 \, \mu M$) for 15 min at $37 \, °C$. Excess TMRM was then removed by centrifugation and platelets resuspended in Tyrode's buffer. Samples were incubated with a combination of CRP and Thrombin (CRP/Thr) at the indicated doses or calcium ionophore A23187 ($1 \, \mu M$) for 10–30 min (where indicated). Samples were diluted into Tyrode's buffer and analysed immediately by flow cytometry.

**Static adhesion.** Glass coverslips were coated with $50 \, \mu g \, ml^{-1}$ human VWF in BSA-free Tyrode's buffer overnight at $4 \, °C$ and blocked with 2% (w/v) BSA for 1 h at room temperature. Washed mouse platelets ($2 \times 10^7 \, ml^{-1}$), were allowed to adhere at $37 \, °C$ in the presence of $10 \, \mu g \, ml^{-1}$ botrocetin. Platelet adhesion was imaged by DIC microscopy (Leica DMIRB, water immersion objective: ×63, numerical aperture: 1.2) and images captured using DVT tools (Pinnacle Systems, USA).

**Platelet aggregation.** Washed mouse platelets ($2 \times 10^8 \, ml^{-1}$) were resuspended in Tyrode's buffer. All aggregation studies[65] were initiated by addition of the indicated concentrations of adenosine diphosphate (ADP) or CRP to platelet suspensions stirred at 600 r.p.m. for 10 min at $37 \, °C$ in a four-channel automated platelet analyser (AggRAM, Helena Laboratories, Tyne and Wear, UK) in the presence of 1 mM calcium and $0.5 \, mg \, ml^{-1}$ fibrinogen. The extent of platelet aggregation was defined as the percentage change in optical density as measured by the automated platelet analyser.

**Quantification of activated partial thromboplastin time.** Activated partial thromboplastin time (APTT) was quantified in a purified system using ellagic acid as a contact pathway activator and a defined number of washed platelets as a source of phospholipid. Briefly, platelet-poor plasma (PPP) was prepared from citrate-anticoagulated whole blood, centrifuged at $2,000g$ for 10 min to separate plasma from blood cells. In a final assay volume of $300 \, \mu l$, PPP ($100 \, \mu l$), ellagic acid (0.1 mM final) and assay buffer (50 mM Hepes, 25 mM Tris, pH 7.35) were pre-warmed ($37 \, °C$) prior to addition to a glass cuvette. Washed human or mouse platelets (resting or activated with CRP/TRAP or CRP/PAR4-AP; $0.5 \, \mu g \, ml^{-1}$ CRP; $1.0 \, \mu M$ TRAP; $200 \, \mu M$ PAR4-AP) were added and the assay mixture incubated at $37 \, °C$ for 2 min. Calcium (25 mM) was finally added to initiate coagulation and APTT measured as the time taken from addition of calcium to the formation of a clot, measured visually by the detection of fibrin strands. In control studies, addition of excess annexin V ($100 \, \mu g \, ml^{-1}$) to the assay prior to initiation of coagulation completely prevented the formation of fibrin. Results were expressed as fold difference over the APTT obtained from control (vehicle-treated human or 14-3-3ζ$^{+/+}$ mouse) platelets.

**Extracellular flux analysis of platelet metabolic function.** The mitochondrial and glycolytic function of isolated platelets was determined through real-time measurement of OCR and ECAR, respectively, using the XFp Extracellular Flux Analyzer (Seahorse Bioscience, MA, USA), according to the manufacturer's instructions. Briefly, washed platelets ($400 \times 10^6 \, ml^{-1}$) were resuspended in Dulbecco's Modified Eagle's Medium (assay medium, Sigma, D5030) containing Hepes (0.02 M, pH 7.35; ICN Biomedicals INC) and apyrase ($0.02 \, U \, ml^{-1}$). In some experiments, washed platelets were pretreated with vehicle or RB-011 ($10 \, \mu M$). Platelets ($20 \times 10^6$ per $200 \, \mu l$) were then loaded into XFp microplate

wells, and maintained at $37 \, °C$ for 30 min, prior to equilibration. For measurement of OCR, assay medium was supplemented with glucose (2.5 M), pyruvate (1 mM) and glutamine (1 mM). OCR was quantified following consecutive treatment of platelets with four different treatments: (1) assay medium alone (resting) or a mix of CRP ($0.25 \, \mu g \, ml^{-1}$) and thrombin ($0.5 \, U \, ml^{-1}$) (stimulated); (2) oligomycin ($1 \, \mu M$); (3) FCCP ($1 \, \mu M$); and (4) rotenone ($5 \, \mu M$) and antimycin A ($5 \, \mu M$). For measurement of ECAR, assay medium was supplemented with Glutamine (1 mM), and ECAR quantified in platelets consecutively treated with (1) assay medium (resting) or a mix of CRP ($0.25 \, \mu g \, ml^{-1}$) and thrombin ($0.5 \, U \, ml^{-1}$; stimulated); (2) glucose (1 M); (3) oligomycin ($1 \, \mu M$); and (4) 2-DG (1 M). For each assay, five individual basal measurements were taken, followed by consecutive injection of treatments. Three measurements were recorded after each injection, with each measurement consisting of 10 s mixing and 3 min measurement period.

**Measurement of metabolic intraplatelet ATP.** To quantify intraplatelet metabolic ATP levels, both total intraplatelet ATP as well as metabolically inert ATP sequestered in platelet granules[66] was quantified, using a modified luciferase assay[67]. Washed mouse platelets ($1 \times 10^8 \, ml^{-1}$) were stimulated with CRP ($0.25 \, \mu g \, ml^{-1}$) and thrombin ($0.5 \, U \, ml^{-1}$) for the indicated times under stirring conditions at $37 \, °C$. In some experiments, washed platelets were pretreated with vehicle or RB-011 ($10 \, \mu M$), prior to stimulation. Assays were stopped at the completion of each time point by the addition of 0.1 M EDTA, pH 7.4. The supernatant containing granular ATP and the pellet containing metabolic ATP were separated via centrifugation. ATP from both fractions was extracted by addition of an ice cold 0.1 M EDTA:Ethanol (9:1) solution containing 1.5 M NaCl, pH 7.4, and separated by centrifugation (13,000 r.p.m., 10 min, $4 \, °C$). A volume of $100 \, \mu l$ of the resultant supernatant was mixed with tris-acetate buffer (0.1 M, 2 mM EDTA, pH 7.75). ATP content was measured by mixing $10 \, \mu l$ of supernatant with $50 \, \mu l$ luciferase/luciferin reagent and luminescence quantified on a fluorescence plate reader (BMG FLUOstar OPTIMA). The ATP content was then calculated using values obtained from an ATP standard curve, and expressed as percentage of control.

**Measurement of glycolytic enzyme activity.** Pyruvate kinase (PK) and 6-phosphofructokinase (6-PFK) activity enzyme activities were quantified from platelet lysates using commercially available colorimetric assay kits (Abcam, Victoria, Australia) according to the manufacturer's instructions, and quantified using an Infinite M1000 Pro micro-plate reader (Tecan, Victoria, Australia). Briefly, washed platelets resuspended in Tyrode's buffer ($1 \times 10^8 \, ml^{-1}$) were treated with buffer alone (resting) or a combination of CRP ($0.25 \, \mu g \, ml^{-1}$) and Thrombin ($0.5 \, U \, ml^{-1}$) (CRP + Thr) for the indicated time periods, followed by the addition of assay-specific extraction buffer. A volume of $10 \, \mu l$ of lysate was diluted with the provided assay buffer, combined with reaction mix and kinetics of enzyme activity quantified over 30 min (25 and $37 \, °C$) at 570 and 450 nm, for PK and 6-PFK, respectively.

**Coimmunoprecipitation of GPIbα and 14-3-3ζ.** Resting platelet whole-cell lysates were prepared using RIPA buffer[65] containing protease (cOmplete) and phosphatase inhibitor (PhosSTOP) cocktails (Roche, Germany). GPIbα was immunoprecipitated from whole-cell lysates (0.2–0.5 mg) using Xia-B2 monoclonal antibody or isotype-matched control immunoglobin G ($5 \, \mu g$) for 1 h, followed by addition of $20 \, \mu l$ protein A/G Plus overnight at $4 \, °C$. SDS–polyacrylamide gel electrophoresis (PAGE) and immunoblot analysis were performed[26,68], following GPIbα immunoprecipitation, whereby samples were mixed with SDS sample buffer in the presence of 0.1 M dithiothreitol, and heated to $95 \, °C$ for 5 min. Equal protein concentration for each sample were subjected to SDS–PAGE on standard 12.5% (v/v) polyacrylamide gel, transferred to polyvinylidene difluoride (PVDF) and probed for GPIb with RAM-6 mAb ($1 \, \mu g \, \mu l^{-1}$; ref. 69) or anti pan-14-3-3 pAb ($0.2 \, \mu g \, ml^{-1}$), followed by horseradish peroxidase (HRP)-conjugated secondary antibody (1:10,000 from the stock solution) and enhanced chemiluminescence.

**SDS–PAGE and western blotting.** Resting platelet whole-cell lysates were prepared as described above. For each sample, equal protein was separated by migration on either standard 12.5% (v/v) polyacrylamide gel (for examination of Phospho-AMPK), or Bio-Rad Criterion XT pre-cast gels (4–12% Bis-Tris; for examination of 14-3-3 isoform expression). Separated proteins were transferred to PVDF membrane and immunoblotted with either Phospho-AMPK (1:1,000), β-actin (1:1,000), 14-3-3α/β (1:1,000), 14-3-3γ (1:1,000), 14-3-3τ (1:1,000) and 14-3-3ζ ($0.4 \, \mu g \, ml^{-1}$) primary antibodies, followed by incubation with either anti-rabbit IgG HRP-conjugated secondary antibody[70] (1:10,000) and enhanced chemiluminescence, for examination of 14-3-3 isoform expression, or IRDye 680LT Donkey anti-Mouse IgG ($0.2 \, \mu g \, ml^{-1}$) and IRDye 800CW Donkey anti-Rabbit IgG ($0.2 \, \mu g \, ml^{-1}$) secondary antibodies for the examination of β-actin and Phospho-AMPK, respectively. Quantification was performed using either a BIORAD ChemiDoc MP 3304 imager (Bio-Rad Laboratories, New South Wales, Australia) or an Odyssey CLx Imager (Li-Cor Biotechnology, Millennium Science, Australia). Original uncropped immunoblots have been provided in Supplementary Fig. 8.

*In vitro* **flow-based thrombus formation.** Glass microslides were coated with Type I fibrillar collagen (250 µg ml$^{-1}$). Hirudin-anticoagulated whole blood was perfused at 1,800 s$^{-1}$ and thrombus formation was monitored for 5 min. Adherent platelets and thrombi were observed using a Leica DMIRB microscope (Leica Microsystems, Wetzlar, Germany) and DIC microscopy ($\times$ 63, 1.2 W objective). Data were recorded onto DVD for offline analysis using Metamorph (Molecular Devices, CA) or ImageJ software (National Institutes of Health).

**Tail bleeding time.** A 3 mm tail-tip transection method was used to assess haemostasis in 20–25 g male and female anaesthetized and ventilated mice[22]. Following transection, the mouse tail was immediately immersed into warmed (37 °C) saline. The bleeding time was determined as the time from the tail transection to the moment the blood flow stopped for more than 120 s. A bleeding time beyond 30 min was considered as the cutoff time for the purpose of statistical analysis. Red blood cells were pelleted and lysed in 1 ml H$_2$0. Haemoglobin was quantified by absorbance at 575 nm (Beckman DU530 Life Science UV/Vis Spectrophotometer). Following cessation of bleeding, the incision was monitored for a further 120 s, and further bleeding within this period classified as 're-bleeding'. The duration of rebleeding was monitored as described above. Non-genotyped mice were used for the 3 mm tail-tip assay, and were subsequently genotyped using tail tissue collected post euthanasia.

**Electrolytic model of thrombosis.** Electrolytic injury of the carotid artery[20] was performed using 14-3-3ζ-deficient, 14-3-3ζ-wt litter mates and bone marrow-transplanted mice, where indicated. Mice were anaesthetized with sodium pentobarbitone (60 mg kg$^{-1}$), and body temperature was monitored/maintained at 37 °C throughout the experiment (using both a heat lamp and thermoblanket (Harvard Apparatus Ltd, Kent, UK). Following surgical exposure of carotid arteries, a calibrated flow probe (0.5 mm i.d.) linked to a flow meter (TS420, Transonic Systems Inc., Ithaca, NY, USA) was placed around the left carotid artery to measure blood flow velocity (in ml min$^{-1}$, finally corrected for body weight (ml min$^{-1}$ per 100 g)). After 10 min of stable blood flow, a hook-shaped electrode was placed around the left carotid artery, distal to the flow probe. Blood stasis was created by clamping the left carotid artery, and vascular injury induced by delivering an electrical current (18 mA, 2 min). Following removal of the clamp, changes in blood flow were monitored for 60 min and recorded for offline analysis using PowerLab LabChart 6 software (ADInstruments, New South Wales, Australia). The total amount of blood flowing through the injured artery following vascular injury was determined by calculating the area under the blood flow curve using Prism software (Version 5.01, Graphpad). Patency time cumulative time during 60 min measurement period that vessel remained open (blood flow > 0 ml min$^{-1}$) and time to first occlusion (blood flow ≤ 0 ml min$^{-1}$) were also analysed. The effect of 14-3-3ζ deficiency on stability of blood flow post injury was classified into three categories: no blood blow (black bar), unstable blood flow (white bar) and stable blood flow (hatched bar). Note: no blood flow was characterized by occlusion of the carotid artery post injury, with no return of blood flow in the 60 min post injury observation period. Unstable blood flow was characterized either by periods of arterial occlusion, followed by periods of returned blood flow, or blood flow that did not occlude but rather demonstrated patterns of decreasing and increasing flow, denoting partial thrombus formation and embolization, respectively. Stable blood flow was characterized by a complete return of blood flow post injury, with no occlusion or fluctuations in the flow.

**Thromboembolism induced by collagen and epinephrine.** A mixture of 1.0 mg kg$^{-1}$ Type I fibrillar collagen and 60 µg kg$^{-1}$ epinephrine were injected into the jugular vein of anaesthetized mice. Blood samples were taken at 10 and 30 min post injection for determination of platelet count and to measure thrombin/anti-thrombin (TAT) complexes[22] as a measure of thrombin generation. Mice were monitored for time of death and lungs were collected, fixed in 10% formalin, and 5 µm sections cut from paraffin-embedded lung tissue. Sections were stained with Masson Trichrome and analysed for the presence of thrombi on an Olympus BX51 microscope; images were captured using a DP70 camera and DP70-BSW software (Olympus) ($\times$ 20 magnification).

**Statistical analysis.** Statistical significance between treatment groups was analysed using either a one- or two-way ANOVA (with Boneforroni's, Sidak's or Tukey's *post hoc* testing, as recommended) or an unpaired Student's *t*-test with two-tailed *P* values (Prism software; GraphPAD Software for Science, San Diego, CA), as indicated. Data are presented as means ± s.e.m., where '*n*' equals the number of independent experiments performed.

**Data availability.** The data that support the findings of this study are available from the corresponding author upon reasonable request.

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

## Acknowledgements

We acknowledge Professor Ben Kile (WEHI), Dr Andre Samson (HRI and University of Sydney), Dr Matthew Mackenzie (Monash MIMR), Dr Sascha C. Hughan and Dr Akiko Ono (ACBD, Monash University) and Mr Carl Coolen (Centre for Cancer Biology, SA Pathology) for advice and technical assistance; Stephen Cody from Monash MicroImaging for help with image analysis; the Monash Histology Platform (Monash University, Clayton Campus, Victoria, Australia), and The Adelaide Proteomics Centre (APC) at the University of Adelaide. This work was supported by funding from the National Health and Medical Research Council (NHMRC) of Australia (Project grant APP1023029—S.M.S.). S.P.J. is the recipient of an NHMRC Senior Principal Research Fellowship. H.S.R. is supported by the Peter Nelson Leukaemia Research Fund. E.E.G. and R.K.A. are both recipients of an NHMRC senior research fellowship. Correspondence relating to 14-3-3ζ-deficient mice and the 14-3-3 dimer destabilizer RB-011 should be addressed to A.F.L. (Centre for Cancer Biology, SA Pathology and University of South Australia, Frome Road, Adelaide SA 5000; Angel.Lopez@health.sa.gov.au).

## Author contributions

S.M.S., R.D. and S.L.C.—planned/performed experiments, analysed data, wrote manuscript. H.S.R.—provided intellectual input, vital reagents and analysed data. S.L.O.—planned/performed experiments and analysed data. S.S.—performed experiments. Y. Yuan—performed experiments and analysed data. Y. Yao—performed experiments. J.R.K.—Planned experiments, analysed data and provided intellectual input. J.W.—provided intellectual input and vital reagents. J.M.—performed experiments and analysed data. S.P.—provided vital reagents. D.C.H.—performed experiments and analysed data. Z.Z.—performed experiments. D.v.d.W.—performed experiments. E.E.G.—performed experiments, analysed data and provided intellectual input.

M.C.B.—provided intellectual input. R.K.A.—provided intellectual input. D.E.J.—provided intellectual input. A.F.L.—provided intellectual input. S.P.J.—planned experiments and wrote manuscript.

## Additional information

**Competing financial interests:** The authors declare no competing financial interests.

DOI: 10.1038/ncomms16125    OPEN

# Corrigendum: 14-3-3ζ regulates the mitochondrial respiratory reserve linked to platelet phosphatidylserine exposure and procoagulant function

Simone M. Schoenwaelder, Roxane Darbousset, Susan L. Cranmer, Hayley S. Ramshaw, Stephanie L. Orive, Sharelle Sturgeon, Yuping Yuan, Yu Yao, James R. Krycer, Joanna Woodcock, Jessica Maclean, Stuart Pitson, Zhaohua Zheng, Darren C. Henstridge, Dianne van der Wal, Elizabeth E. Gardiner, Michael C. Berndt, Robert K. Andrews, David E. James, Angel F. Lopez & Shaun P. Jackson

Nature Communications 7:12862 doi: 10.1038/ncomms12862 (2016); Published 27 Sep 2016; Updated 30 Aug 2017

In Supplementary Fig. 7 of this Article, graphs presenting 6-PFK kinetics in panel d were inadvertently duplicated from those in panel c. The correct version of Supplementary Fig. 7 appears below as Fig. 1.

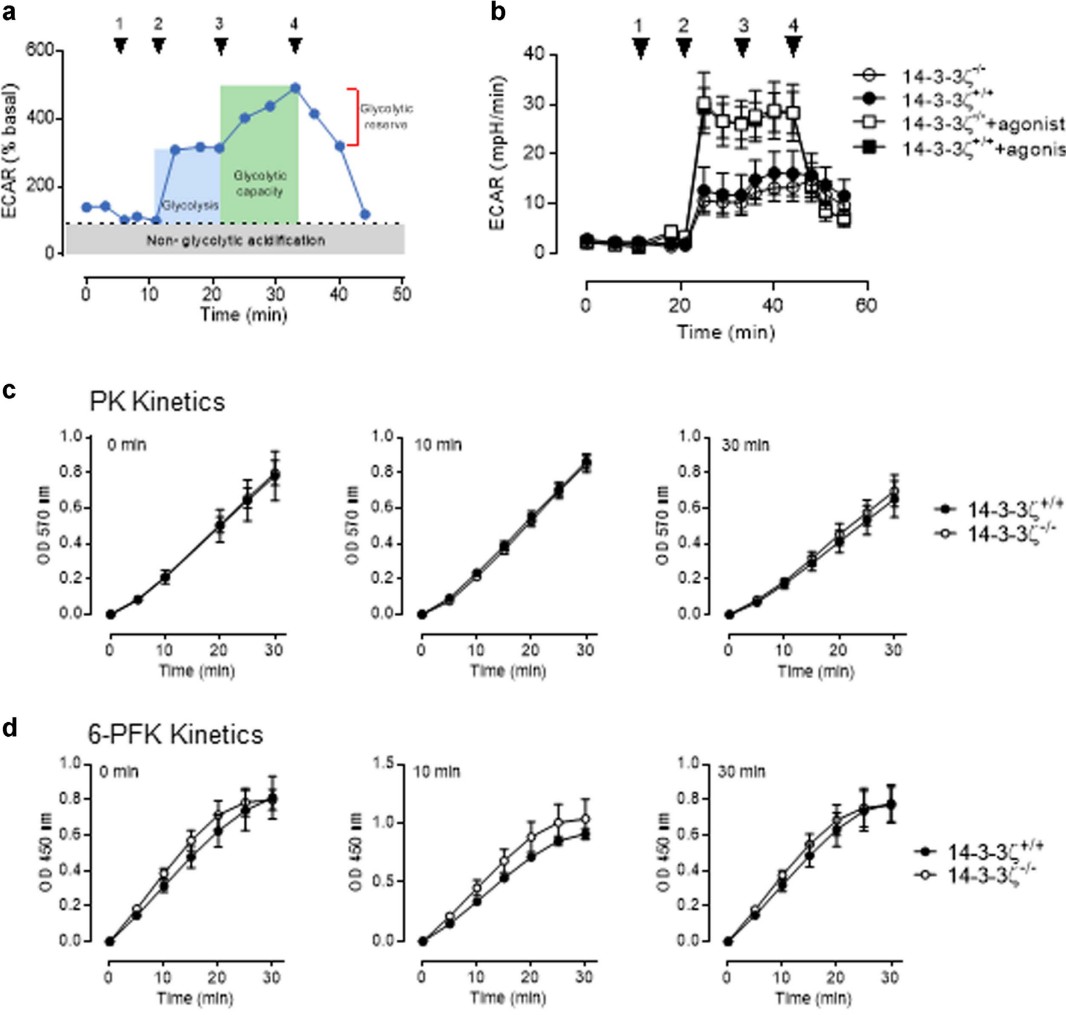

**Figure 1**

