## [Peer review file · Nature Communications]

Reviewers' comments:

Reviewer #1 (expert in platelets biology and thrombosis)

Remarks to the Author:

This manuscript by Schoenwaeder et al entitled "14-3-3ζ regulates the mitochondrial respiratory reserve linked to platelet phosphatidylserine exposure and procoagulant activity" explores the function of 14-3-3ζ in platelets and concludes that this protein plays an important role in the expression of phosphatidylserine during platelet activation. This manuscript is comprehensive and highly detailed, and poses a new conceptual approach to understanding the expression of a procoagulant surface during thrombin formation. Overall, it is comprehensive but tedious given the large number of bar graphs and detailed text. However, it is conceptually novel and addresses an important issue. Using 14-3-3ζ null mice, the authors compare platelet activation with these and wild type mice using diverse but standard methods.

Comments:

1. Everything possible should be done to contract the text by judicious editing. The large number of bar graph, both main and supplementary, is distracting. Is all of this necessary? On some figures (e.g. Figure 3), the subfigures are so small that the writing on these figures can not be read. The number of supplementary figures is out of proportion to that of other papers.
2. Annexin V is used to determine the presence of phosphatidylserine expression on platelet membrane surfaces. Although this is a routine application, how specific is annexin V for phosphatylserine membrane bilays? Can annexin V bind to other membrane surfaces. This paper is very dependent on annexin V reporting PS.
3. In the absence of any meaningful physiologic data, is it of value to suggest that 14-3-3ζ might be a target for antithrombotic therapy? It seems that 50% of papers in this field make such suggestions-and are not further evaluated.
4. P 21, l 18. "Data" is plural. Should be "was."

Reviewer #2 (expert in platelets biology)

Remarks to the Author:

This is a very well-executed study that provides convincing evidence that 14-3-3(zeta) is critical for platelet phosphatidylserine (PS) exposure and what is called the pro-coagulant response. The authors used mice lacking 14-3-3(zeta) specifically in megakaryocytes/platelets to investigate how this adaptor protein contributes to platelet function. To their surprise, they did not find evidence for a critical role of 14-3-3(zeta) in GPIb signaling, as suggested by several recent studies. Instead, they provide convincing evidence that 14-3-3(zeta) is critical for the platelet pro-coagulant response. Consistent with the defect in PS exposure, 14-3-3(zeta) knockout mice were protected from thrombosis without affecting hemostasis. On a molecular level, the authors show that 14-3-3(zeta) knockout platelets exhibit better bioenergetics upon strong cellular stimulation. A mechanistic explanation for this effect was not provided.

Concerns:

- 1) The work by Jobe et al. (Blood 2008) in cyclophilin D-deficient animals showed that a defect in the formation of the mitochondrial permeability transition pore leads to markedly reduced PS exposure BUT an increased risk of thrombosis. On a molecular level, this was explained by the inability of cypD-KO platelets to downregulate integrin affinity in procoagulant platelets. How do the authors explain the anti-thrombotic effect of 14-3-3(zeta)-deficiency in light of the findings in cypD-KO mice? Why was integrin activation in 14-3-3(zeta) KO platelets not studied upon activation by CRP/PAR4p (supplemental Fig. 4)? Supplemental Figure 7 shows no difference in integrin activation in human platelets activated with CRP/PAR4p in the presence and absence of the 14-3-3 inhibitor?!
- 2) PS exposure should be studied in blood perfused over collagen under venous and arterial flow conditions. Does lack of 14-3-3(zeta) also affect ballooning?
- 3) Figure 3d: the y-axis labels are confusing. Since CRP/TRAP stimulation leads to a shortening of the aPTT, shouldn't the "fold shortening" be a number greater than 1?
- 4) Figure 4: what is the effect of anticoagulation on the collagen/epinephrine model? Are heparinized mice even more protected than 14-3-3(zeta) KOs? Also, there is no difference in the platelet count between control and 14-3-3(zeta) KO mice - wouldn't that be expected if thrombin generation contributes to the thrombotic phenotype?

Reviewer #3 (expert in platelets biology and signaling)

Remarks to the Author:

In this manuscript, the authors describe their analysis of phosphatidylserine exposure in platelets from wild type and 14-3-3 knockout mice. The authors show that 14-3-3 is involved in regulating mitochondrial function and phosphatidylserine exposure and thus platelet procoagulant activity. This is a well written and elegant body of work however there are some weaknesses in the current manuscript.

Figure 4 shows the effect of 14-3-3 deficiency on collagen/epinephrine induce thromboembolism. It is unclear if the histological images were from mice that survived (particularly the 14-3-3^{+/+} mice). It would be interesting to compare lungs from mice that lived versus those that died for each genotype.

One page 9 of the text, in reference to Fig. 4d, the authors state "without impacting platelet count". As written this implies to the reader that there was no change in platelet count compared to controls whereas there was a large decrease in platelet count but no differences between the genotypes. This should be clarified.

There are no obvious thrombi in the lungs of the 14-3-3^{-/-} mice however the platelet count drops to a similar extent to the 14-3-3^{+/+} mice. The authors should comment on this result.

The data presented in Fig. 6 show annexin V binding and mitochondrial membrane potential. Is the reduction in mitochondrial membrane potential in the 14-3-3^{+/+} and 14-3-3^{-/-} platelets statistically significant at 10 minutes and 30 minutes compared to the zero time point? Dual labeling studies (annexin V and TMRM) should be carried out to assess how the level of mitochondrial membrane potential depolarization equates to annexin V binding.

Minor:

The y-axis labels for graphs showing flow cytometry data are "(% gated)" it is unclear what this means and should be clarified

The 14-3-3 uncoupling agent RB-011 was used to show that 14-3-3 proteins are involved in phosphatidylserine expression and procoagulant activity. It would be useful to assess how this compound affects intraplatelet metabolic ATP (figure 5).

Reviewer #4 (expert in 14-3-3 proteins)

Remarks to the Author:

The authors here examine whether 14-3-3 ζ can regulate platelet function. They find that 14-3-3 ζ has an isoform-specific function in regulating platelet procoagulant function by regulating mitochondrial bioenergetics and phosphatidylserine exposure. Overall, the study is well done and the results are described clearly. The interpretations and conclusions are justified by their experiments. Some comments are outlined below of some revisions that need to be considered to improve the manuscript:

- 1) The description of the statistical analysis is limited, and per the methods section, the authors state that they used Dunnett's test for post-hoc testing. Dunnett's test is only appropriate for comparing several conditions to only one group (typically control). However, they make multiple intergroup comparisons involving several groups and not just to one control group. In this scenario, they should use Tukey's, Newman-Keuls, Sidak, or Bonferroni's tests, which are found in Graphpad Prism (which is the program they report that they used for statistical analyses).
- 2) In Fig. 4b, they show micrographs of lung sections to demonstrate that 14-3-3 ζ knockout mice show less development of pulmonary embolism in their model. Clot burden in these mice should be quantified if possible.
- 3) In Supp. Fig 5, the authors show 2D gel of 14-3-3 expression from platelet lysates from 14-3-3 ζ +/+ and knockout mice. They should label each band as to which 14-3-3 isoform is represented.
- 4) In Supp. Fig 6b, the authors show co-immunoprecipitation results for 14-3-3 and GPIIb. They state that there is no reduction in 14-3-3s co-immunoprecipitating with GPIIb in knockout platelets, but there does seem to be some potential difference between wildtype and knockout platelets which may be more apparent in a blot that is not overexposed. They need to show a blot with decreased exposure and potentially adjust their conclusion.
- 5) In the discussion section (p. 14), the authors state that this is the first demonstration that 14-3-3 regulates mitochondrial reserve capacity. Their data suggests that 14-3-3 ζ promotes bioenergetic failure leading to cell necrosis and ultimately PS exposure and clot formation. However, there is a body of literature that 14-3-3s can protect against mitochondrial toxins and mitochondria-mediated apoptosis (for example, *J. Cell Physiol* 2011 226:2329-37; *PLoS One* 2011 6:e21720; *Cell Signal*. 2015; 27:770-6, among others). How does their data fit in with these other known effects of 14-3-3s on mitochondria and cell survival? In general, 14-3-3s promote cell survival, and their results suggest that 14-3-3 ζ instead may promote cell death as part of its regulation of platelet function. This should be addressed in the discussion section.

Response to reviewer comments

NCOMMS-16-01210

We would like to thank all reviewers for their thorough review of our studies and for their constructive suggestions.

Reviewers' comments:

Reviewer #1 (expert in platelet biology and thrombosis)

Comments:

1. Everything possible should be done to contract the text by judicious editing. The large number of bar graph, both main and supplementary, is distracting. Is all of this necessary? On some figures (e.g. Figure 3), the subfigures are so small that the writing on these figures cannot be read. The number of supplementary figures is out of proportion to that of other papers.

We agree with the reviewer that there is a large amount of data included in “*Supplementary Information*”. We have consolidated 3 of the *Supplementary Figures* into a combined “*Figure 3*” in the main manuscript. In doing so, we have reduced the number of *Supplementary Figures* from 10 to 7, whilst the main figure count has increased to 8. We have not removed any further *Supplementary Information* in order to remain compliant with the ***Nature Communications*** policy that states “*avoid ‘data not shown’ statements and instead include data necessary to evaluate the claims of the paper as Supplementary Information*” (<http://www.nature.com/ncomms/authors/submit.html#General-info>).

2. Annexin V is used to determine the presence of phosphatidylserine expression on platelet membrane surfaces. Although this is a routine application, how specific is annexin V for phosphatylserine membrane bilays? Can annexin V bind to other membrane surfaces. This paper is very dependent on annexin V reporting PS.

Annexin V is a benchmark probe for assessment of PS exposure (Koopman *et al*, *Blood*, 84(5):1415-1420, 1994; Vermes *et al*, *J Immunol Methods*. 184(1):39-51, 1995). Lactadherin is also used for this purpose and binds PS with high sensitivity and specificity (Dasgupta *et al*, *Transl Res.*, 148(1):19-25, 2006; Shi *et al*, *Cytometry A*, 69(12):1193-201, 2007; Albanyan *et al*, *Transfusion*. 2009 Jan;49(1):99-107). Notably, annexin V binding to cellular membranes requires Ca^{2+} and is enhanced by the presence of phosphatidylethanolamine¹, whereas lactadherin binding is not affected by Ca^{2+} nor the presence of phosphatidylethanolamine².

We have performed experiments in human platelets using labelled lactadherin. Similar to our findings with annexin Va, there is minimal PS exposure (lactadherin binding) on the surface of resting platelets, and a time-dependent increase in lactadherin binding following potent stimulation of platelets with CRP/thrombin. Pretreating platelets with the 14-3-3 inhibitor RB-011 resulted in a marked reduction in lactadherin binding, identical to our findings with annexin Va (*Figure 1 – for review purposes only* [FIGURE REDACTED]), confirming a role for these adaptor proteins in regulating platelet PS expression. We have modified the “*Results*” section accordingly (Page 8, second paragraph, lines 1-2) to indicate that similar findings were obtained with FITC-Lactadherin.

3. In the absence of any meaningful physiologic data, is it of value to suggest that 14-3-3 ζ might be a target for antithrombotic therapy? It seems that 50% of papers in this field make such suggestions-and are not further evaluated.

The physiological data relevant to this issue is bleeding. Our studies using a standard tail bleeding time, as well as analysis of surgical bleeding, have revealed that 14-3-3 ζ deficiency is not associated with an obvious bleeding tendency (Fig. 2e), despite good protection from both carotid artery thrombosis (Fig. 1; Fig. 2 a-d) and fatal pulmonary embolism (Fig. 5). These findings are consistent with a growing body of evidence that reducing the procoagulant function of platelets may reduce propagation of thrombi, without necessarily undermining the hemostatic response of platelets^{3,4}. We agree with the reviewer that further detailed studies will be required to fully evaluate the likely antithrombotic efficacy and potential safety of inhibiting 14-3-3 ζ in platelets. Nonetheless, we think it is reasonable to conclude at this stage that 14-3-3 ζ may be an interesting therapeutic target to reduce thrombosis, particularly in the context of cancer therapy, where dysregulated 14-3-3 signaling is common. This issue has been further elaborated upon in the final paragraph of the “discussion” (Pages 15-16).

4. P 21, l 18. "Data" is plural. Should be "was."

We have made this correction in the text. Note: this change appears on page 22 of the revised manuscript, last paragraph, line 5.

Reviewer #2 (expert in platelet biology)

Remarks to the Author:

A mechanistic explanation for this effect was not provided.

Defining the precise biomechanical mechanisms by which 14-3-3 signaling proteins regulate specific cellular responses is challenging, in large part because 14-3-3 family members interact with such a large repertoire of intracellular molecules, including signaling proteins, metabolic enzymes, cytoskeletal proteins, transcription factors, apoptosis regulators and tumor suppressor proteins. To date, there is limited evidence that any single 14-3-3-protein interaction can independently induce a specific cellular response.

Our finding of dysregulated ATP levels in 14-3-3 ζ deficient platelets has indicated an important role for 14-3-3 ζ in regulating cell metabolism following potent platelet stimulation. However, the precise molecular mechanisms by which 14-3-3 ζ regulates cellular metabolic function remains incompletely understood, despite evidence that 14-3-3 proteins can regulate a host of glycolytic and mitochondrial enzymes. We have found no evidence of dysregulated glycolysis in 14-3-3 ζ deficient platelets, based on the analysis of known 14-3-3-regulated glycolytic enzymes and from extracellular flux analysis (Suppl. Fig. 7). It is more likely that 14-3-3 ζ plays an important role in regulating mitochondrial energy metabolism since analysis of mitochondrial bioenergetics (oxygen consumption rate) revealed an enhanced respiratory reserve capacity in 14-3-3 ζ ^{-/-} platelets (Fig 8). We have examined the possibility that 14-3-3 ζ may regulate the mitochondrial F₀F₁ ATP synthase, as this enzyme plays a major role in regulating ATP production and cell survival. Consistent with this, we have shown that oligomycin (an ATP synthase inhibitor) negated the difference observed in 14-3-3 ζ ^{+/+} and 14-3-3 ζ ^{-/-} platelets, with respect to PS exposure and metabolic ATP (Fig. 6, 7). However, we think it is likely that 14-3-3 ζ also regulates other mitochondrial processes linked to ATP generation, since 14-3-3 ζ ^{-/-} platelets exhibited enhanced respiratory reserve capacity even when the ATP synthase is inhibited with oligomycin (Fig 8).

We acknowledge that a considerable amount of additional work will be required to unravel the complex interactions between 14-3-3 ζ and the various effector proteins involved

in regulating the mitochondrial respiratory reserve. As this represents a poorly defined area of cell metabolism in general, we expect that unravelling this process will not be straightforward, and in all likelihood, will represent a major body of work in its own right.

Concerns:

1) The work by Jobe et al. (Blood 2008) in cyclophilin D-deficient animals showed that a defect in the formation of the mitochondrial permeability transition pore leads to markedly reduced PS exposure BUT an increased risk of thrombosis. On a molecular level, this was explained by the inability of cypD-KO platelets to downregulate integrin affinity in procoagulant platelets. How do the authors explain the anti-thrombotic effect of 14-3-3(zeta)-deficiency in light of the findings in cypD-KO mice?

Reviewer 2 has raised an interesting issue, which has been investigated by several groups, with somewhat conflicting data^{5,6}. As outlined by the reviewer, Jobe and colleagues demonstrated reduced PS exposure in *CypD*^{-/-} platelets *in vitro*, that was associated with a prothrombotic phenotype in a photochemical injury model of carotid artery thrombosis⁶. From our own experience with this model, the level of platelet activation is quite low (low level P-selectin expression and minimal levels of platelet PS exposure), hence it is unclear whether platelet-dependent procoagulant function plays an important role in promoting thrombus formation in this model. Unfortunately, this issue wasn't addressed in the manuscript by Jobe *et al.*,

In contrast, Hua and colleagues have recently reported a decrease in procoagulant platelets in cyclophilin D-deficient mice that is associated with a decreased thrombotic response in both FeCl₃ and laser injury models⁵. These authors confirmed that platelet PS exposure was prominent in the FeCl₃ injury model. This suggests that the impact of cyclophilin D deficiency on thrombosis is context dependent, varying considerably between different experimental models. It is also possible that methodological differences between these studies may be important, as the former study by Jobe used global *CypD*^{-/-} mice, whereas Hua and colleagues used PF4-Cre conditional *CypD* knockout mice, hence it is possible that non-platelet related effects of *CypD* deficiency may have altered the thrombotic response reported by Jobe *et al.* Interestingly, in our own unpublished studies with PF4-Cre conditional *CypD* knockout mice we have seen a mild antithrombotic phenotype in an electrolytic model of carotid thrombosis (*Figure 2 – for review purposes only* [FIGURE REDACTED]), which provides further support for the concept that the impact of *CypD* deficiency on the thrombotic response is highly dependent on the experimental model used.

Why was integrin activation in 14-3-3(zeta) KO platelets not studied upon activation by CRP/PAR4p (supplemental Fig. 4)?

The main aim of the studies reported in the original *Supplementary figure 4* (revised – *Fig. 3e, f*) was to examine whether standard platelet activation responses are preserved in 14-3-3 ζ ^{-/-} platelets, as a means of understanding the prothrombotic defect observed *in vivo*. We quantified JON/A binding (integrin $\alpha_{IIb}\beta_3$ activation) in 14-3-3 ζ ^{+/+} and 14-3-3 ζ ^{-/-} platelets and confirmed normal integrin $\alpha_{IIb}\beta_3$ activation in these mice. We have also examined CRP/PAR4 activated human platelets in the absence or presence of the 14-3-3 dimer destabiliser (*Suppl. Fig. 3a, b*) and revealed no significant defect in integrin $\alpha_{IIb}\beta_3$ activation or P-selectin expression.

Supplemental Figure 7 shows no difference in integrin activation in human platelets activated with CRP/PAR4p in the presence and absence of the 14-3-3 inhibitor?!

These findings are not surprising, as the decline in integrin affinity is time-dependent and also influenced by the potency of platelet stimulation. At the time points and agonist concentrations used in the studies depicted in Supplementary Figure 3 (originally *Suppl. Fig. 7*), there was no significant decrease in integrin activation in either 14-3-3 $\zeta^{+/+}$ or 14-3-3 $\zeta^{-/-}$ platelets. In contrast, studies performed with calcium ionophore (*Suppl. Fig. 3b*), which is a much more potent inducer of PS exposure and procoagulant function, greatly reduced integrin activation, a finding consistent with the studies of Jobe⁶.

2) PS exposure should be studied in blood perfused over collagen under venous and arterial flow conditions. Does lack of 14-3-3(zeta) also affect ballooning?

As requested by this reviewer, we have performed *in vitro* perfusion studies to examine the number of PS+ve ballooning platelets forming under flow conditions (300 s⁻¹) during adhesion of washed platelets to Type I collagen (250 μ g/ml). These studies have demonstrated a reduction in the number of platelets adopting a procoagulant balloon morphology in blood treated with the 14-3-3 dimer destabiliser (RB-011), when compared with untreated platelets. These results have been quantified and presented as a histogram in *Supplementary Figure 3c*.

3) Figure 3d: the y-axis labels are confusing. Since CRP/TRAP stimulation leads to a shortening of the aPTT, shouldn't the "fold shortening" be a number greater than 1?

We apologise for our use of confusing nomenclature. The scatter plot provided in our original Fig. 3d (*now in revised Figure 4d*) was presented as aPTT relative to resting platelets, which were allocated a value of 1.0. This figure demonstrates shortening of aPTT following activation with CRP/Thr, a shortening which is significantly negated in the presence of the 14-3-3 dimer destabiliser (RB-011). The y-axis is labelled as "APTT (*relative to resting*)".

4) (i) Figure 4: what is the effect of anticoagulation on the collagen/epinephrine model? Are heparinized mice even more protected than 14-3-3(zeta) KOs? (ii) Also, there is no difference in the platelet count between control and 14-3-3(zeta) KO mice - wouldn't that be expected if thrombin generation contributes to the thrombotic phenotype?

The collagen/epinephrine model of pulmonary embolism has been well characterised and is both a platelet- and coagulation-dependent model⁷⁻¹¹. It is most commonly used in the assessment of the prothrombotic function of platelets^{7,9}, although it has also been used to determine the role of contact factor activation of blood coagulation in pulmonary thromboembolism^{8,10} as well as in the assessment of fibrinolytic agents¹².

We have confirmed by histology that 14-3-3 ζ deficiency is associated with a significant decrease in thrombus burden in the pulmonary circulation following administration of collagen and epinephrine (see *Fig. 5b, ii*). Notably, 14-3-3 $\zeta^{-/-}$ mice demonstrated a comparable drop in circulating platelet count to their wild type counterparts (*Fig. 5d*). Similar findings were demonstrated across 2 different genetic strains of mice (BalbC and sv129). These findings suggest that the protection conferred in this model by 14-3-3 ζ -deficiency is not due to a defect in collagen binding or presumably subsequent platelet aggregation, as large platelet aggregates could be observed by histology. Rather, our interpretation is that 14-3-3 ζ deficiency leads to reduced thrombin generation, a finding consistent with decreased TAT levels in these mice. Consistent with such a possibility are previous findings on FXII deficient mice¹³. These mice are protected from lethal pulmonary

embolism despite normal platelet activation. As with our 14-3-3 mice, the marked reductions in platelet count in the FXII deficient mice was similar to wildtype controls¹³. A reduction in thrombin generation in these models and concurrent reduced fibrin generation may render platelet aggregates more unstable and less likely to occlude vessels. In support of this, we have evidence of reduced clot stability in 14-3-3 ζ deficient mice following carotid artery electrolytic injury (*Fig. 1a, e*). We have modified the manuscript (*Results*: page 5, paragraph 1; page 9, paragraph 1) to clarify this point.

We have preliminary evidence that high dose hirudin (5mg/kg) prevents occlusive thrombus formation in our collagen-epinephrine mouse model, without preventing severe thrombocytopenia, similar to the findings in FXII-deficient mice¹³. However, these hirudin doses are far beyond normal therapeutic levels (aPPT increased 2-3 fold at 0.1-0.25mg/kg) and we will need to perform more detailed dose-response studies to determine the relative protection from thrombosis in 14-3-3 ζ deficient mice relative to therapeutically relevant doses of heparin or hirudin. We believe these additional studies are beyond the scope of the current manuscript.

Reviewer #3 (expert in platelets biology and signaling) Remarks to the Author:

Figure 4 shows the effect of 14-3-3 deficiency on collagen/epinephrine induce thromboembolism. It is unclear if the histological images were from mice that survived (particularly the 14-3-3^{+/+} mice). It would be interesting to compare lungs from mice that lived versus those that died for each genotype.

In our *original figure 4 (revised Fig. 5)*, the Masson's Trichrome-stained lung section taken from a 14-3-3 ζ ^{+/+} mouse demonstrated extensive occlusive platelet thrombi (as denoted by yellow arrows). This particular mouse died at 10 minutes, with a platelet count of 0. As stated in the legend, the extensive thrombi observed in this mouse was absent in surviving 14-3-3 ζ ^{-/-} mice, with a histological section provided from one of the surviving 14-3-3 ζ ^{-/-} mice (platelet count at 30 min = 20 x10³/ μ l). It should be noted however that platelet aggregates can be observed in KO sections (these have been denoted in the figure with arrows). We have provided additional information in the legend to *figure 5* to clarify these points for the reader.

We have also provided additional histology (*Figure 3 - for review purposes only [FIGURE REDACTED]*), to allow further comparison of lung sections taken from mice from both genotypes - varied in their survival and platelet count. **(a)** Example of a 14-3-3 ζ ^{+/+} mouse (#535) that survived at 30'00" post-agonist administration, with evidence of more extensive thrombi compared to surviving 14-3-3 ζ ^{-/-} mouse (#533). **(b)** Example of mice from both genotypes that had similar platelet counts at 10'00", however while the 14-3-3 ζ ^{+/+} mice died at 14'26" (#564), the corresponding 14-3-3 ζ ^{-/-} mouse survived to 30'00" (#605).

One page 9 of the text, in reference to Fig. 4d, the authors state "without impacting platelet count". As written this implies to the reader that there was no change in platelet count compared to controls whereas there was a large decrease in platelet count but no differences between the genotypes. This should be clarified.

We agree with the reviewer that the text as written could be misinterpreted. We have therefore clarified this statement on page 9 to include the following: "significantly lower levels of TAT in 14-3-3 ζ ^{-/-} mice both at 10 and 30 min post-challenge, despite a similar drop in platelet count in 14-3-3 ζ ^{-/-} mice relative to 14-3-3 ζ ^{+/+} mice."

There are no obvious thrombi in the lungs of the 14-3-3^{-/-} mice however the platelet count drops to a similar extent to the 14-3-3^{+/+} mice. The authors should comment on this result.

Please see response to Reviewer 2, question 4. We have modified the manuscript (Results, page 8, final paragraph; page 9, paragraph 1; Figure 5, legend) accordingly to clarify this point.

The data presented in Fig. 6 show annexin V binding and mitochondrial membrane potential. Is the reduction in mitochondrial membrane potential in the 14-3-3^{+/+} and 14-3-3^{-/-} platelets statistically significant at 10 minutes and 30 minutes compared to the 'zero' time point? Dual labelling studies (annexin V and TMRM) should be carried out to assess how the level of mitochondrial membrane potential depolarization equates to annexin V binding.

We have re-analysed the statistical significance of 14-3-3 $\zeta^{+/+}$ and 14-3-3 $\zeta^{-/-}$ platelets at 10 minutes and 30 minutes, compared to each of their respective 'zero' time points. This re-analysis is provided in *Figure 4(a) – for review purposes only* [FIGURE REDACTED]. In 14-3-3 $\zeta^{+/+}$ mice, there is statistical significance in the level of mitochondrial membrane depolarisation ($\Delta\psi_m$) following both 10 and 30 minutes of stimulation with CRP/Thr, with p values <0.01 (***) and <0.001 (****), respectively. In contrast, 14-3-3 $\zeta^{-/-}$ deficient mice demonstrate no statistical significant loss in $\Delta\psi_m$ at 10 minutes, with significance reached at 30 minutes, with a p-value of <0.05 (*). These results are consistent with 14-3-3 $\zeta^{-/-}$ deficient mice demonstrating less PS exposure.

We have also performed single- and dual labelling studies with Annexin V and TMRM, and have observed a consistent decrease in mitochondrial membrane potential ($\Delta\psi_m$) in conjunction with an increase in PS exposure (similar to that observed in single-labelling studies), confirming a correlation between these 2 events in platelets stimulated with potent agonists. A representative experiment depicting one such dual-labelling experiment is presented in *Figure 4(b,c) – for review purposes only* [FIGURE REDACTED].

Minor:

The y-axis labels for graphs showing flow cytometry data are "(% gated)" it is unclear what this means and should be clarified.

When analysing platelet populations by flow cytometry, the purified platelet population was first gated, and then analysed for positive events (Positive for Annexin V binding). Resting platelets were considered as a negative control, expressing <2% events positive for Annexin V binding. Annexin V positivity of the platelet population was therefore presented as a % of the gated platelet population, and compared to the resting population.

The 14-3-3 uncoupling agent RB-011 was used to show that 14-3-3 proteins are involved in phosphatidylserine expression and procoagulant activity. It would be useful to assess how this compound affects intraplatelet metabolic ATP (figure 5).

To address this issue, we have performed additional studies in human platelets pretreated with vehicle or RB-011. Assessment of the impact of this inhibitor both on metabolic ATP depletion following activation, as well as OCR using Extracellular flux analysis have revealed similar results to those obtained comparing 14-3-3 $\zeta^{+/+}$ and 14-3-3 $\zeta^{-/-}$ mouse platelets. Specifically, these studies have demonstrated partial protection from metabolic depletion in response to stimulation of platelets with the potent agonist combination CRP/thrombin, as well as a potentiation of reserve capacity in RB-011-pretreated CRP/thrombin stimulated platelets. We have included these additional results in *Supplementary Figure 6*, and have

adjusted the “Results” to include this additional information accordingly (Page 10, lines 4-5; page 12, lines 7-9).

Reviewer #4 (expert in 14-3-3 proteins)

Remarks to the Author:

1) The description of the statistical analysis is limited, and per the methods section, the authors state that they used Dunnett's test for post-hoc testing. Dunnett's test is only appropriate for comparing several conditions to only one group (typically control). However, they make multiple intergroup comparisons involving several groups and not just to one control group. In this scenario, they should use Tukey's, Newman-Keuls, Sidak, or Bonferroni's tests, which are found in Graphpad Prism (which is the program they report that they used for statistical analyses).

Under the ‘Methods’ we have modified our description of the statistical analysis to include all relevant tests used (page 24). Accordingly, and where relevant, we have also re-analysed various data sets, in line with the comments of this reviewer, to include appropriate post-hoc testing (as indicated in corresponding figure legends).

In summary:

Main Figure 1: Un-paired student t-test

Main Figure 2: Un-paired student t-test

Main Figure 3: Un-paired student t-test; (c) 2-way ANOVA (Bonferroni's post-hoc); (d) 1-way ANOVA (Tukey's post-hoc); (d, inset, e) Un-paired student t-test; (f) 1-way ANOVA (Tukey's post-hoc)

Main Figure 4: (c) 1-way ANOVA (Bonferroni's post-hoc)

Main Figure 5: (a) 2-way ANOVA (tukey's post-hoc); (c) Un-paired student t-test

Main Figure 6: (a,b) 2-way ANOVA (Bonferroni's post-hoc)

Main Figure 7: (b) 2-way ANOVA (Sidak's post-hoc); (c) Un-paired student t-test.

Suppl Figure 1: (a,b) 2-way ANOVA (Bonferroni's post-hoc);

Supp Figure 2: (b) Un-paired student t-test;

Suppl Figure 3: (c) 2-way ANOVA (Bonferroni's post-hoc);

Suppl. Figure 4: 2-way ANOVA (Bonferroni's post-hoc);

Suppl. Figure 7: 2-way ANOVA (Bonferroni's post-hoc);

Suppl. Figure 8: 2-way ANOVA (Bonferroni's post-hoc);

Suppl. Figure 9: 2-way ANOVA (Bonferroni's post-hoc);

Suppl. Figure 10: 2-way ANOVA (Bonferroni's post-hoc).

2) In Fig. 4b, they show micrographs of lung sections to demonstrate that 14-3-3ζ knockout mice show less development of pulmonary embolism in their model. Clot burden in these mice should be quantified if possible.

Clot burden was quantified using histological sections taken from mice subjected to the pulmonary embolism model, described under “Methods”. The number of vessels with occlusive platelet-rich thrombi were quantified from each section (field of view), with at least 3-4 fields of view assessed per mouse, with 4 mice analysed per genotype (n=4). These results, which have been added to Figure 5b (Fig 5b, ii), demonstrate at least a 2-3 fold reduction in clot burden in 14-3-3ζ deficient mice compared to wild type littermates. Please also refer to response to Reviewer v#2, 4(i).

3) In Supp. Fig 5, the authors show 2D gel of 14-3-3 expression from platelet lysates from 14-3-3ζ +/- and knockout mice. They should label each band as to which 14-3-3 isoform is represented.

We have repeated immunoblot analysis using isoform-specific antibodies against 14-3-3 β/α , γ , ζ and τ . 14-3-3 β , γ and ζ are the predominant isoforms in platelets, and these studies have confirmed the complete loss of 14-3-3 ζ from 14-3-3 $\zeta^{-/-}$ platelets. These studies also indicate some upregulation of 14-3-3 γ in 14-3-3 ζ deficient platelets. This information is provided in a revised *Figure 3g*, with “*Results*” modified accordingly (Page 7, paragraph 1).

4) In Supp. Fig 6b, the authors show co-immunoprecipitation results for 14-3-3 and GPIb α . They state that there is no reduction in 14-3-3s co-immunoprecipitating with GPIb α in knockout platelets, but there does seem to be some potential difference between wildtype and knockout platelets which may be more apparent in a blot that is not overexposed. They need to show a blot with decreased exposure and potentially adjust their conclusion.

As requested, we have performed additional studies on 14-3-3 association with GPIb α . Immunoprecipitated GPIb α from platelet lysates was immunoblotted with a pan-selective anti-14-3-3 polyclonal antibody (Santa Cruz Biotechnology, K19). A low exposure immunoblot confirms that there was no major reduction in the level of 14-3-3 protein binding to GPIb α (Fig. 3i) in 14-3-3 $\zeta^{-/-}$ platelets relative to 14-3-3 $\zeta^{+/+}$ controls. Whether this is partially due to upregulation of the 14-3-3 γ isoform remains to be determined. We have accordingly modified the text; P7 line 9-13.

5) In the discussion section (p. 14), the authors state that this is the first demonstration that 14-3-3 regulates mitochondrial reserve capacity. Their data suggests that 14-3-3 ζ promotes bioenergetic failure leading to cell necrosis and ultimately PS exposure and clot formation. However, there is a body of literature that 14-3-3s can protect against mitochondrial toxins and mitochondria-mediated apoptosis (for example, J. Cell Physiol 2011 226:2329-37; PLoS One 2011 6:e21720; Cell Signal. 2015; 27:770-6, among others). How does their data fit in with these other known effects of 14-3-3s on mitochondria and cell survival? In general, 14-3-3s promote cell survival, and their results suggest that 14-3-3 ζ instead may promote cell death as part of its regulation of platelet function. This should be addressed in the discussion section.

We agree with the reviewer that the role of 14-3-3 ζ (and other 14-3-3 isoforms) in regulating cell death is well defined. This is most clearly demonstrated in the context of apoptosis and to a lesser extent, autophagy. To date, there are no reports of 14-3-3 adaptor proteins modulating regulated cell necrosis, and it was completely unexpected that 14-3-3 ζ would promote platelet bioenergetic failure and PS exposure. We could find no evidence that 14-3-3 ζ influences platelet apoptosis, based on our findings with the BH3 mimetic ABT-737 (Suppl. Fig. 4a) and through the demonstration of a normal platelet lifespan in 14-3-3 $\zeta^{-/-}$ platelets (Suppl. Fig. 1a). We have modified the discussion (page 14) as underlined below.

“Much speculation has surrounded the signaling mechanisms leading to development of platelet procoagulant function, with a growing body of evidence supporting a central role for mitochondrial-driven cell death pathways in this process¹⁵. Two cell death pathways can induce platelet PS exposure and procoagulant function - programmed cell apoptosis and regulated cell necrosis¹⁶. The former pathway plays a central role in regulating the lifespan of circulating quiescent platelets¹⁷, which we show here is unaffected in 14-3-3 $\zeta^{-/-}$ platelets. Interestingly, our own studies using mouse platelets deficient in the proapoptotic Bcl-2 family proteins Bak and Bax have revealed that development of activation-dependent PS exposure is unperturbed, instead demonstrating that necrotic-like cell death pathways are principally involved in regulating agonist-induced development of procoagulant platelets¹⁶. Cell necrosis is typically caused by bioenergetic failure of the cell and the studies presented in

this manuscript are consistent with this concept, demonstrating that reduced platelet PS exposure in 14-3-3 ζ deficient platelets correlates with enhanced mitochondrial bioenergetic function, sustained levels of metabolic ATP and reduced AMPK activation. To our knowledge there are no reports of 14-3-3 adaptor proteins influencing regulated cell necrosis pathways linked to platelet bioenergetic function and PS exposure.”

References

1. Swairjo, M.A., Concha, N.O., Kaetzel, M.A., Dedman, J.R. & Seaton, B.A. Ca²⁺-bridging mechanism and phospholipid head group recognition in the membrane-binding protein annexin V. *Nat Struct Biol* **2**, 968-974 (1995).
2. Shi, J. & Gilbert, G.E. Lactadherin inhibits enzyme complexes of blood coagulation by competing for phospholipid-binding sites. *Blood* **101**, 2628-2636 (2003).
3. Braun, A., *et al.* Orai1 (CRACM1) is the platelet SOC channel and essential for pathological thrombus formation. *Blood* **113**, 2056-2063 (2009).
4. Varga-Szabo, D., Braun, A. & Nieswandt, B. STIM and Orai in platelet function. *Cell calcium* **50**, 270-278 (2011).
5. Hua, V.M., *et al.* Necrotic platelets provide a procoagulant surface during thrombosis. *Blood* **126**, 2852-2862 (2015).
6. Jobe, S.M., *et al.* Critical role for the mitochondrial permeability transition pore and cyclophilin D in platelet activation and thrombosis. *Blood* **111**, 1257-1265 (2008).
7. Mizurini, D.M., Francischetti, I.M. & Monteiro, R.Q. Aegyptin inhibits collagen-induced coagulation activation in vitro and thromboembolism in vivo. *Biochemical and biophysical research communications* **436**, 235-239 (2013).
8. Decrem, Y., *et al.* Ir-CPI, a coagulation contact phase inhibitor from the tick *Ixodes ricinus*, inhibits thrombus formation without impairing hemostasis. *J Exp Med* **206**, 2381-2395 (2009).
9. He, L., *et al.* The contributions of the alpha 2 beta 1 integrin to vascular thrombosis in vivo. *Blood* **102**, 3652-3657 (2003).
10. Ma, D., *et al.* Desmolaris, a novel factor Xla anticoagulant from the salivary gland of the vampire bat (*Desmodus rotundus*) inhibits inflammation and thrombosis in vivo. *Blood* **122**, 4094-4106 (2013).
11. Mackman, N. Triggers, targets and treatments for thrombosis. *Nature* **451**, 914-918 (2008).
12. Radnaabazar, C., Park, C.M., Kim, J.H., Cha, J. & Song, Y.S. Fibrinolytic and antiplatelet aggregation properties of a recombinant Cheonggukjang kinase. *J Med Food* **14**, 625-629 (2011).
13. Renne, T., *et al.* Defective thrombus formation in mice lacking coagulation factor XII. *J Exp Med* **202**, 271-281 (2005).
14. Mangin, P.H., *et al.* Identification of five novel 14-3-3 isoforms interacting with the GPIb-IX complex in platelets. *J Thromb Haemost* **7**, 1550-1555 (2009).
15. Jackson, S.P. & Schoenwaelder, S.M. Procoagulant platelets: are they necrotic? *Blood* **116**, 2011-2018 (2010).
16. Schoenwaelder, S.M., *et al.* Two distinct pathways regulate platelet phosphatidylserine exposure and procoagulant function. *Blood* **114**, 663-666 (2009).
17. Mason, K.D., *et al.* Programmed anuclear cell death delimits platelet life span. *Cell* **128**, 1173-1186 (2007).

REVIEWERS' COMMENTS:

Reviewer #1 (Remarks to the Author):

This is a revision of a manuscript previously submitted to Nature Communications from Dr. S. Jackson and his group. They have addressed some of my issues and in particular the lacadherin experiment is nicely. The remaining suggestions that I made were of an editorial nature and I defer to the opinions of the editorial staff.

I still believe that this paper, as comprehensive as it is, is too long for its message, and the total number of figures (true total) is too great. But this is a single opinion. Similarly the boiler plate statement about the potential of this adaptor protein as a target for antithrombotic therapy is very speculative given limited duration of animal experiments and does not deserve more than a sentence. I could argue that the fact that this protein is involved in so many cellular processes that the chance of success is very small. But you never know.

I will defer to the signalling experts more familiar with this protein.

Reviewer #2 (Remarks to the Author):

no further comments

Reviewer #3 (Remarks to the Author):

In this revised manuscript, the authors describe their analysis of phosphatidylserine exposure in platelets from wild type and 14-3-3zeta knockout mice. The authors show that 14-3-3 is involved in regulating mitochondrial function and phosphatidylserine exposure and thus platelet procoagulant activity. They also demonstrate that the 14-3-3 destabilizing agent RB-011 mimics the loss of 14-3-3. This is a well written and elegant body of work and the authors have adequately addressed this reviewers initial concerns.

Minor:

The x-axis label in Supp Fig 3a, and Fig 4b are CRP + Thr whereas all other labels are CRP/Thr.

Reviewer #4 (Remarks to the Author):

The authors here examine whether 14-3-3ζ can regulate platelet function. They find that 14-3-3ζ has an isoform-specific function in regulating platelet procoagulant function by regulating mitochondrial bioenergetics and phosphatidylserine exposure. Overall, the study is well done and the results are described clearly. The interpretations and conclusions are justified by their experiments. The authors have addressed most of the concerns of the reviewers. With regard to the specific criticisms brought up by this reviewer, the authors have addressed most of the concerns. Some minor issues still need to be addressed:

- 1) The authors have better explained their statistical approach and have adjusted most of their multiple intergroup comparisons to appropriate testing. However, in figure 2, their figure legend suggests that for 2b-d they used unpaired t-test. This is not an appropriate test as more than 2 groups are in the experiment and an ANOVA followed by post-hoc test is more appropriate.
- 2) The authors compare the two genotypes with regard to 14-3-3 isoform expression as requested. They report that there is an increase in 14-3-3gamma expression in the knockout, but their blots also seem to suggest an increase in 14-3-3tau/theta isoform also (Fig. 3g). Also, they state that there is a reduction in pan14-3-3 signal, but this is not apparent in the blot they show in

Fig 3g. The increases in the gamma and theta isoforms could compensate for the loss in zeta to maintain total pan 14-3-3 levels. Please amend the comments on p7 in the results section regarding these blots (lines 162-165) or else show quantification of the blots to support the current comments.

Response to reviewer comments

NCOMMS-16-01210B

We would like to thank all reviewers for their review of our revised manuscript. In the text below, we have outlined changes made to the manuscript in response to requests from Reviewers 3 and 4.

Reviewer #3

Request:

Minor:

The x-axis label in Supp Fig 3a, and Fig 4b are CRP + Thr whereas all other labels are CRP/Thr.

We have altered the x-axis label in both figures (Supplementary 3a and Figure 4b) such that both read “CRP/Thr”, consistent with the remaining manuscript.

Reviewer #4

Remarks to the Author:

1) The authors have better explained their statistical approach and have adjusted most of their multiple intergroup comparisons to appropriate testing. However, in figure 2, their figure legend suggests that for 2b-d they used unpaired t-test. This is not an appropriate test as more than 2 groups are in the experiment and an ANOVA followed by post-hoc test is more appropriate.

We apologise for this oversight. We have re-analysed these figures in Fig. 2, using one-way ANOVA with Sidak’s post-hoc testing. The appropriate p values have been confirmed in the figure, and the correct statistical test has been referred to in the legend for Figure 2.

2) The authors compare the two genotypes with regard to 14-3-3 isoform expression as requested. They report that there is an increase in 14-3-3gamma expression in the knockout, but their blots also seem to suggest an increase in 14-3-3tau/theta isoform also (Fig. 3g). Also, they state that there is a reduction in pan14-3-3 signal, but this is not apparent in the blot they show in Fig 3g. The increases in the gamma and theta isoforms could compensate for the loss in zeta to maintain total pan 14-3-3 levels. Please amend the comments on p7 in the results section regarding these blots (lines 162-165) or else show quantification of the blots to support the current comments.

As suggested, we have removed the speculative statements regarding the increases in other isoforms (gamma, tau/theta), compensating for loss of 14-3-3ζ. This modified text appears on page 7 (and can be seen in tracked changes).